# Unifying the design space and optimizing linear and nonlinear truss metamaterials by generative modeling

Li Zheng [1], Konstantinos Karapiperis [1], Siddhant Kumar [2] ✉ & Dennis M. Kochmann [1] ✉

The rise of machine learning has fueled the discovery of new materials and, especially, metamaterials—truss lattices being their most prominent class. While their tailorable properties have been explored extensively, the design of truss-based metamaterials has remained highly limited and often heuristic, due to the vast, discrete design space and the lack of a comprehensive parameterization. We here present a graph-based deep learning generative framework, which combines a variational autoencoder and a property predictor, to construct a reduced, continuous latent representation covering an enormous range of trusses. This unified latent space allows for the fast generation of new designs through simple operations (e.g., traversing the latent space or interpolating between structures). We further demonstrate an optimization framework for the inverse design of trusses with customized mechanical properties in both the linear and nonlinear regimes, including designs exhibiting exceptionally stiff, auxetic, pentamode-like, and tailored nonlinear behaviors. This generative model can predict manufacturable (and counterintuitive) designs with extreme target properties beyond the training domain.

Architected metamaterials are rapidly redefining the boundaries of achievable material properties. Supported by additive manufacturing, the design of such cellular solids with tailored microstructural architecture has led to unprecedented functionality: from counter-intuitive negative compressibility[1,2] and negative Poisson's ratio[3,4] to mechanical cloaking[5], extreme energy absorption[6–8], and guided acoustic waves[9,10]. Among the myriad of available design spaces, truss metamaterials-based on periodic lattices of beam networks—have emerged as the dominant one, particularly due to their high stiffness and strength in the ultralow-relative-density regime[11–16] and the simple manufacturability.

Truss metamaterials offer an extensively tunable design space based on both the lattice topology (i.e., the connectivity of the beam network) as well as geometric features (e.g., the length, orientation, and cross-sectional shape of each strut). However, most of this unlimited design freedom has remained untapped. Many design applications[17–21] have been limited to a small catalog of ad-hoc lattices (e.g., kagome,

octet, and octahedron[22–28]), which have been identified through a combination of intuition and trial-and-error over the years. While the catalog-based search space can be enriched by tuning geometric features[29] or base material properties[19], it is strongly limited in topological tunability and fails to exploit the full range of achievable designs and hence of achievable effective metamaterial properties. Many truss optimization solutions have adopted heuristic search strategies to find optimal structures by iteratively adjusting the active beams and/or nodes in the design domain, according to mechanics-based criteria[30–33]. This, however, becomes computationally infeasible in large-scale problems due to the enormous and noisy search space. Recently, Lumpe and Stankovic[34] proposed an extensive catalog of truss lattices by mimicking the molecular structure of crystalline lattices. Yet, the same fundamental issue persists as for all such catalogs of truss lattices with different topologies: there exists no finite-dimensional, continuous, and seamless design space. For example, while the "kagome" vs. "octet"

[1]Mechanics & Materials Lab, Department of Mechanical and Process Engineering, ETH Zürich, 8092 Zürich, Switzerland. [2]Department of Materials Science and Engineering, Delft University of Technology, 2628 CD Delft, Netherlands. ✉e-mail: sid.kumar@tudelft.nl; dmk@ethz.ch

lattice is interpretable by humans, it is not directly cognizant to a computer. The question is hence: how does one translate distinct lattice topologies into a unified, finite-dimensional, vector-based parameterization that can be understood by an algorithm aiming to optimize the lattice design for certain target metamaterial properties? While a pixeled/voxelated image-based parameterization (similar to conventional topology optimization) is a solution in principle, capturing slender beams in truss lattices warrants extremely high resolution, which again becomes prohibitively expensive.

To address the aforementioned limitations, we introduce a graph-theoretic approach to represent a vast design space of three-dimensional (3D) truss topologies. Every truss lattice can be naturally translated into a graph—a mathematical structure consisting of edges and nodes (i.e., struts and their intersections, respectively). The edges encode the lattice topology in the form of a nodal adjacency matrix; the nodes encode the geometric features in the form of spatial coordinates. Additional graph-level information may include, e.g., strut thickness or further cross-sectional information. While a graph as a data structure is computationally interpretable, the discontinuities across different lattice topologies are also persistent in the graph representation.

We introduce a machine learning (ML) framework to extract a generalizable and unified design space for truss lattices with diverse topologies. ML has made a significant impact in the design of metamaterials—from data-driven surrogate models for accelerating multi-scale simulations[35–40] to the inverse design for tailored linear[41–47] and nonlinear[48–51] properties. Of particular interest to our context are generative ML models (using, e.g., variational autoencoders[52] (VAEs) and generative adversarial networks[53] (GANs)), which aim to learn the underlying distribution of the data itself (as opposed to discriminative models that learn to predict labels for a given input) and have been used to successfully design metamaterials[21,54–56]. However, unlike in those approaches where the design parameterization can be formulated as a finite-dimensional vector or image, we turn to the special class of generative graph-based ML models for dealing with graph representations of truss lattices. Graph-based learning has recently gained prominence because of its ability to model non-Euclidean data representing interrelations in irregular domains, such as social networks[57–59], chemical molecules[60,61], and material microstructures[37,62]. Distinct from existing works that utilize supervised graph-based models as surrogate models to provide real-time prediction of various properties of interest, e.g., homogenized elastic[63] and thermal properties[64] or dominant deformation mechanism[65] of lattice architectures, the goal of the graph generative modeling framework proposed here is to construct a unified, continuous latent representation of a vast and discrete truss design space and its exploitation for the inverse design for both linear and nonlinear targeted mechanical properties. We here demonstrate that a VAE can successfully abstract a hidden or latent design representation of diverse graph-based truss lattices. This is achieved by using a neural network architecture, which contains an informational bottleneck and enables compressing the high-dimensional graph representation into a finite, low-dimensional, and smooth vector representation. In this latent space, any two lattices with similar topological and geometric features are located close to each other, whereas any two distant lattices can be continuously transformed into each other. New lattice designs can be straightforwardly generated by randomly sampling in the latent space. Exploration of this latent space further allows us to seamlessly search or optimize for truss lattices with exotic or tailored properties—including those that lie outside the domain of the available training data.

## Results

### Creating the design space

We begin by introducing our definition of the design space of truss lattices. From a practical standpoint, we focus our attention on lattices based on the periodic tessellation of a cubic representative volume element (RVE). Inspired by the cube decomposition approach[66], we partition the RVE into eight equal cubes, the octants, as shown in Fig. 1a. By assuming symmetries across the three mutually orthogonal symmetry planes, it is sufficient to define the truss only within a single octant, which simplifies the complex optimization problem while ensuring great flexibility and periodic tilability. Following the graph representation of molecules[63,67], truss-like structures can be described by a set of nodes connected by solid beams, which form the nodes and edges of the graph, respectively. To create a sufficiently large design space of truss structures, we define a total of 27 possible node placements within the octant (Fig. 1a): 8 vertex nodes $\{v_0, v_1, v_2, v_3, v_4, v_5, v_6, v_7\}$, 12 edge nodes $\{e_0, e_1, e_2, e_3, e_4, e_5, e_6, e_7, e_8, e_9, e_{10}, e_{11}\}$, 6 face nodes $\{f_0, f_1, f_2, f_3, f_4, f_5\}$, and a single body node $\{t_0\}$ within the volume. While the body node is free to move in 3D, edge and face nodes are restricted to be displaced only along the edge and within the face, respectively, and vertex nodes are fixed, as illustrated in Fig. 1a. We define the offsets of nodes (as in ref. 66) in the natural coordinate system, representing their relative positions with respect to the fixed vertex nodes to ensure connectivity on the outer boundaries. The complete set of *node features* $x$ contains the offset of each node in its movable direction(s).

Analogous to the above node features, the *structural features* of the unit cell include the truss topology, which is fully described by a list of all edges (e.g., $(e_0, f_1)$ represents a beam connecting edge node $e_0$ and face node $f_1$). By analogy with a graph structure, we represent the lattice topology using an *adjacency matrix* $A \in \{0, 1\}^{n \times n}$, where the diagonal elements $A_{ii} = 1$ (for all $i = 1, \ldots, n$) and $n = 27$ is the total number of nodes. The adjacency matrix serves as a lookup table, where the value of 1 denotes an edge between nodes, whereas 0 indicates that an edge is not present. With the graph representation, introducing or removing beams from the truss can be easily achieved by operations on the adjacency matrix (e.g., the superposition of two structures is described by the logical disjunction, i.e., element-wise boolean OR, of two adjacency matrices). Possible defects, such as isolated nodes or struts, can be efficiently identified and resolved by examining the adjacency matrix. Here we do not explicitly consider permutation invariance or equivariance of graphs due to the inherent representation complexity in generative modeling tasks[68,69]. In other scenarios (such as predicting frame-indifferent properties[37,70]), incorporating permutation invariance or symmetry groups such as SE(3) in graph representations[71–74] could largely enhance the learning of the underlying relations of various structural configurations.

We leverage the above representation to construct a dataset containing a large family of truss lattices covering a wide range of mechanical properties. To this end, we begin with a set of three well-known elementary trusses as initial topologies, including the octet, body-centered-cubic, and simple cubic unit cells as $1 \times 1 \times 1$ and $2 \times 2 \times 2$ tessellations. Starting from those, an iterative stochastic perturbation algorithm generates novel structures by randomly inserting/removing both nodes (of edge, vertex, face, or body types) and truss connectivities. New connections are created by connecting newly inserted nodes to at least one of their nearest neighbor nodes. Random perturbations are added to the position of all nodes, while obeying the corresponding positional constraints of the vertex, edge, and face nodes. The above procedure is repeated until the obtained truss lattice satisfies the constraints on the graph connectivity (details provided in Supplementary Note 1). We build a large preliminary library, which includes a variety of trusses—from well-studied structures to unconventional ones, as shown in Fig. 1c. From the thus-obtained preliminary library, new lattices are created by superimposing two randomly sampled structures (with repetitions allowed). To ensure the physical feasibility of structures generated by random perturbations and superpositions, we enforce the constraint that the lattice is self-connected; i.e., the truss graph must have exactly one maximally

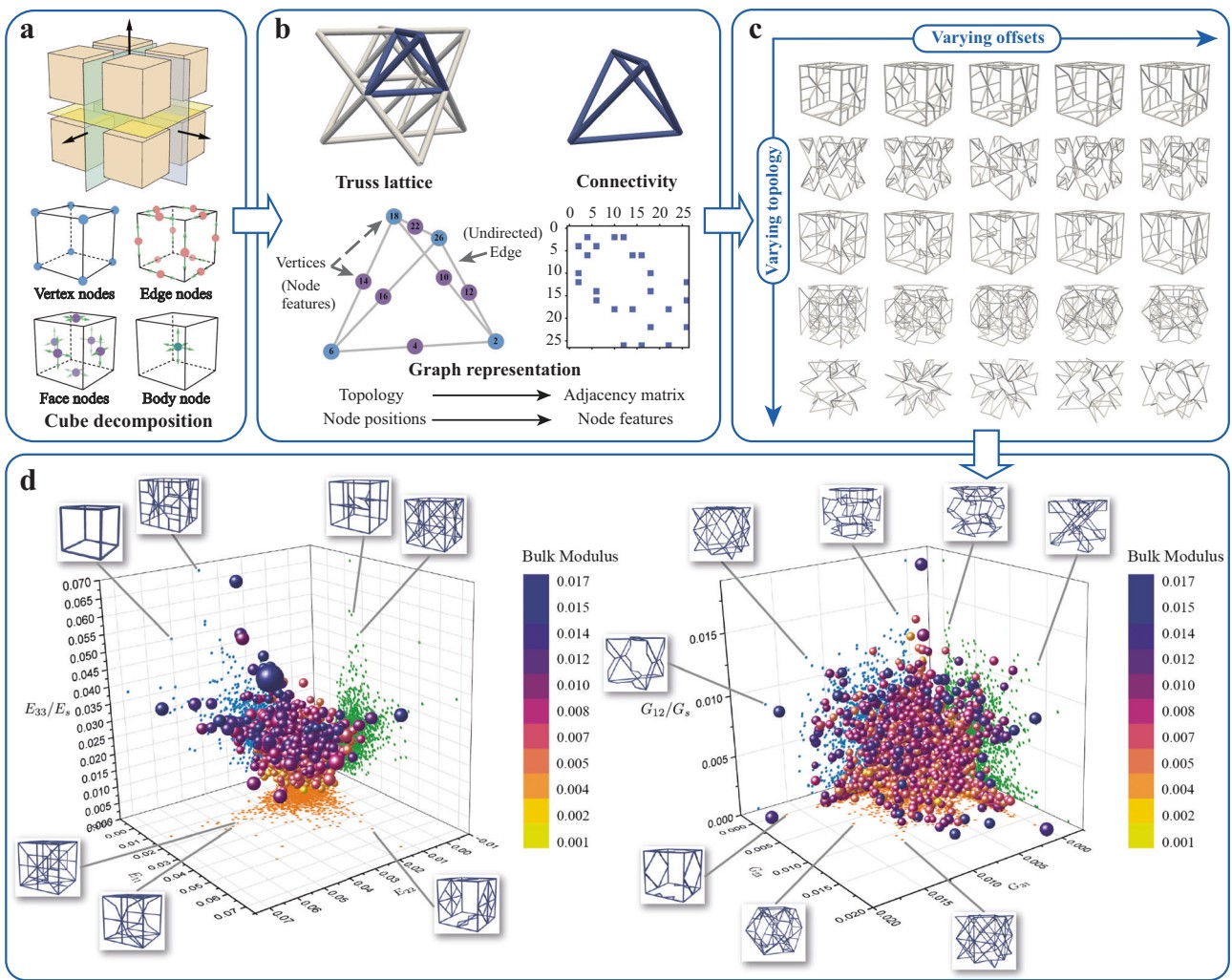

**Fig. 1 | Overview of the truss parametrization and the data generation scheme used to create a diverse truss lattice dataset. a** Cube decomposition generates the irreducible truss pattern with possible node placements and the nodes' degrees of freedom defined on the octant. For example, an edge node has one degree of freedom and is only allowed to traverse along the edge, while a face node with two degrees of freedom can assume any position within the plane. **b** Graph representation of an octet truss example (vertex nodes in blue colors, and face node in purple colors), whose vertices and beams serve as input to the variational auto-encoder (VAE) model. The lattice is defined by the adjacency matrix and node features. **c** Examples of different truss lattices are realized by varying the topology and the vertex degrees of freedom as well as the strut diameters, showcasing the wide coverage of the design space. **d** Effective directional Young's moduli $E$ and effective shear moduli $G$ (normalized by the respective properties $E_s$ and $G_s$ of the base material) in the three main directions and their projected values on the $\boldsymbol{e}_1$–$\boldsymbol{e}_2$-, $\boldsymbol{e}_1$–$\boldsymbol{e}_3$-, and $\boldsymbol{e}_2$–$\boldsymbol{e}_3$-planes of 3000 structures randomly drawn from the dataset, and selected examples with extreme properties. The size of the markers is proportional to the strut radius of the unit cell; their colors indicate the effective bulk modulus. The effective stiffness was obtained by finite element (FE) homogenization with periodic boundary conditions; the radius of circular cylindrical struts is scaled to maintain a constant relative density of $\rho = 0.15$. The resulting variety of truss configurations in the dataset covers a large range of elastic properties.

connected subgraph spanning the whole graph. Intersecting beams, which may arise from the superposition of different topologies, are fixed by splitting the involved beams and inserting a new node at the intersection point. The resulting lattices constitute the design space as well as the corresponding dataset for ML.

By perturbing both the topological and geometrical features of the lattice, we create a rich database of anisotropic lattices, consisting of 965,736 unique structures. As a representative and important mechanical property, we study the full anisotropic 3D elastic stiffness tensor. All lattices obtained from the cube decomposition approach naturally possess three orthogonal plane reflection symmetries and therefore only require nine independent components to describe the orthotropic homogenized stiffness tensor; we select $\boldsymbol{S} = (\mathbb{C}_{1111},$ $\mathbb{C}_{1122}, \mathbb{C}_{1133}, \mathbb{C}_{2222}, \mathbb{C}_{2233}, \mathbb{C}_{3333}, \mathbb{C}_{2323}, \mathbb{C}_{3131}, \mathbb{C}_{1212})$. For each structure, the effective mechanical stiffness tensor is computed by homogenization, using a finite element (FE) framework, which models individual struts as linear elastic Timoshenko beams with a circular cross-section[75]. We

assume a homogeneous base material with Poisson's ratio $\nu_s = 0.3$ and unit Young's modulus $E_s = 1$ (i.e., all reported effective stiffness values are relative to the base material's Young's modulus). The beam thickness $d$ of all struts is varied such that a constant relative density $\rho = 0.15$ is maintained across all structures. This will be helpful during property optimization, as it ensures that optimized mechanical properties do not come at the cost of an increased weight.

To visualize the property range reached by the established truss catalog, Fig. 1d shows the effective directional Young's moduli $E$ and the effective shear moduli $G$ along the three principle cubic directions and their projections onto the x-y-, x-z-, and y-z-planes. Results show that the established truss database covers a wide range of Young's moduli, spanning three orders of magnitude between $10^{-5}$ and $10^{-2}$ times the base material's Young's modulus. Although the initial structures used as seeds are limited to cubic symmetry, the resulting library generated by perturbing both the truss connectivity and node positions exhibits strong anisotropy and has significantly

expanded the range of mechanical properties (e.g., the representative examples shown in Fig. 1d reach effective Young's moduli $E_{33}$ of ca. 36% higher than that of a simple cubic unit cell in the principle direction at the same density). Of course, the dataset could be enriched by more unique structures, using the above approach. Yet, we limit our study to the current dataset based on the performance and computational cost of the ML model (as detailed below). The data generation could also be generalized to other truss families. For example, while we only consider centrosymmetric unit cells, non-centrosymmetric unit cells can be readily constructed in a similar manner by allowing for different structures in each octant instead of applying symmetries. Compared to prior approaches that have focused on a truss unit cell catalog, our database achieves a significantly wider design space of truss structures with a relatively compact formulation based on graphs.

## Generative modeling framework

The proposed design space for truss lattices is discrete and discontinuous. For example, any two truss lattices may have different numbers of nodes or nodes with different numbers of degrees of freedom (such as edge vs. face nodes). However, representing the truss lattices by a graph structure, as described above, enables the use of node features and adjacency matrix as inputs for a ML model to learn a low-dimensional, continuous, and smooth representation for the high-dimensional, discrete, and intractable graph representation.

We use a VAE containing two neural networks–an *encoder* and a *decoder* (see Fig. 2a for a schematic of the framework). Let $G = (A, x)$ denote the graph representation of a truss lattice, defined by its adjacency matrix and node features. The encoder $\mathcal{Q}_\phi$ (with the set of trainable parameters $\phi$) maps an input graph $G$ into two $d$-dimensional vectors $\mu(G; \phi) \in \mathbb{R}^d$ and $\sigma(G; \phi) \in \mathbb{R}^d$, which, respectively, prescribe the mean and covariance of a diagonal multivariate Gaussian distribution

$$z \sim \mathcal{N}\left([\mu_1, \ldots, \mu_d]^\top, \operatorname{diag}\left([\sigma_1^2, \ldots, \sigma_d^2]^\top\right)\right). \quad (1)$$

Here, $z \in \mathbb{R}^d$ denotes a low-dimensional vector encoding of the input graph $G$, also known as latent representation. While the formulation is presented in terms of $\sigma$, the neural network predicts $\log \sigma$. To maintain differentiability (required for backpropagation-based neural network training), $z$ is sampled using the reparameterization trick[52] as

$$z = \mu + \varepsilon \odot [\sigma_1, \ldots, \sigma_d]^\top \quad \text{with} \quad \varepsilon \sim \mathcal{N}(0, I), \quad (2)$$

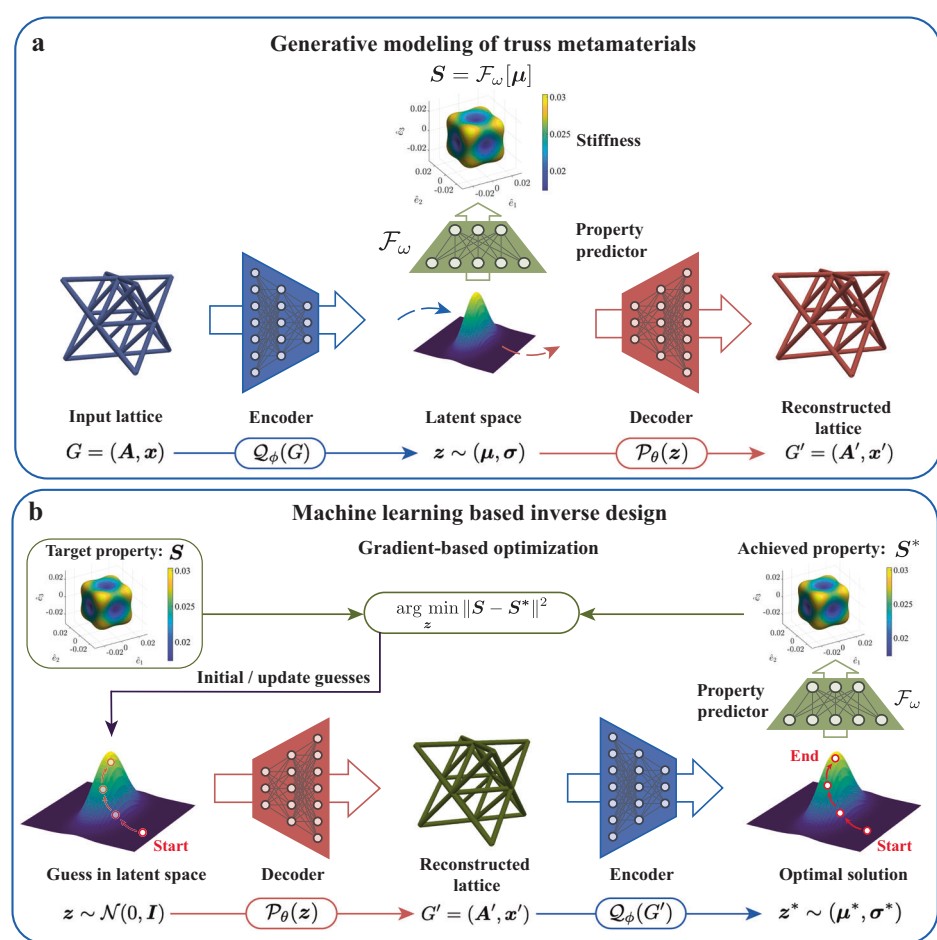

**Fig. 2 | Generative modeling framework. a** The variational autoencoder (VAE) model takes the graph representation $G = (A, x)$ of truss lattices (defined by adjacency matrix $A$ and node features $x$) as input to the encoder $\mathcal{Q}_\phi$ and learns a continuous latent space over the geometries of various trusses. $\mu$ and $\sigma$ denote the mean and covariance of the multivariate Gaussian distribution. The reduced representation $z$ of truss structures is then passed to the decoder $\mathcal{P}_\theta$ to reconstruct the lattice. An augmented multi-layer perceptron (MLP) neural network $\mathcal{F}_\omega$ predicts the mechanical properties of trusses based on their latent representation $z$. **b** The inverse design framework aims to generate truss lattices with target properties. Starting with the 100 closest matches in the training dataset as initial guesses, gradient-based optimization is applied to search for possible lattices with desirable properties in the latent space ($\mathcal{N}(0, I)$ denotes a uniform distribution of zero mean and the identity as the covariance matrix). The inverse design candidate structures are passed to the encoder to obtain their corresponding latent representations, which are then forwarded to the property predictor to predict the effective stiffness of the proposed lattices.

where $\odot$ denotes element-wise multiplication. The decoder $\mathcal{P}_\theta$ (with set of trainable parameters $\theta$) maps the latent vector $\boldsymbol{z}$ into a graph representation $G' = (\boldsymbol{A}', \boldsymbol{x}') = \mathcal{P}_\theta(\boldsymbol{z}; \theta)$ and attempts to accurately reconstruct the original graph, i.e., $G \approx G'$. The autoencoding of the input graphs with such a neural network structure creates an information bottleneck[76] in the latent representation. The information bottleneck only preserves the necessary meaningful information to allow accurate reconstruction of the graphs, with a significant reduction in the dimension and complexity of the original data. Consequently, in the latent space any two graphs/lattices with similar topology and geometry are located close to each other; any two distant graphs/lattices can be continuously transformed between each other by traversing the latent space.

We here adopt the attributed network embedding method[77] to learn the individual dependencies of the structural topology and node placements as well as their combined effects (details provided in Supplementary Note 2.2). Specifically, the adjacency matrix (containing binary values) and node features (containing continuous values) are serialized and passed through separate encoders to obtain the respective latent space distribution means $\boldsymbol{\mu}^A \in \mathbb{R}^{d_A}$ and $\boldsymbol{\mu}^X \in \mathbb{R}^{d_X}$ and standard deviations $\boldsymbol{\sigma}^A \in \mathbb{R}^{d_A}$ and $\boldsymbol{\sigma}^X \in \mathbb{R}^{d_X}$. The embedding dimensions $d_A$ and $d_X$ are chosen such that $d_{Ax} = (d_A + d_X) - d > 0$. Note that, since the adjacency matrix is symmetric, only the upper triangular part is considered by the encoder. The final latent space distribution is obtained by partial overlap of the adjacency matrix and node features embeddings. The mean is given by

$$\boldsymbol{\mu} = \underbrace{\begin{bmatrix} \mu_1^A \\ \vdots \\ \mu_{d_A - d_{Ax}}^A \end{bmatrix}}_{\text{topology-specific}} \oplus \underbrace{\frac{1}{2}\left( \begin{bmatrix} \mu_{d_A - d_{Ax}+1}^A \\ \vdots \\ \mu_{d_A}^A \end{bmatrix} + \begin{bmatrix} \mu_1^X \\ \vdots \\ \mu_{d_{Ax}}^X \end{bmatrix} \right)}_{\text{shared topology and geometry}} \oplus \underbrace{\begin{bmatrix} \mu_{d_{Ax}+1}^X \\ \vdots \\ \mu_{d_X}^X \end{bmatrix}}_{\text{topology-specific}} , \quad (3)$$

where $\oplus$ denotes vector concatenation (the logarithm of the standard deviation, i.e., $\log \boldsymbol{\sigma}$, is obtained analogously using $\log \boldsymbol{\sigma}^A$ and $\log \boldsymbol{\sigma}^X$). Since the adjacency matrix and node features influence the topology and geometry, respectively, subsets of the latent space dimensions offer topology-specific, geometry-specific, and shared control over the design space, the advantages of which will become apparent when discussing the results. Similar to the encoding, two separate decoders are used to output the graph $G' = (\boldsymbol{A}', \boldsymbol{x}')$ from a latent vector $\boldsymbol{z}$–the topology-specific and shared dimensions of $\boldsymbol{z}$ are used to obtain the adjacency matrix $\boldsymbol{A}'$; the shared and geometry-specific dimensions of $\boldsymbol{z}$ are used to obtain the node features $\boldsymbol{x}'$.

Towards the efficient data-driven design and the discovery of new structures with desirable properties, the latent space can be associated with specific properties that we seek to optimize by a neural network surrogate model[21,78] using the latent vectors as input, thus bypassing the costly FE homogenization computation. Therefore, we adapt the original VAE structure and link the latent space to the homogenized effective stiffness measures $\boldsymbol{S}$ by feeding the mean of the latent vector, i.e., $\boldsymbol{\mu}(G; \phi)$, into an additional neural network-based property predictor $\mathcal{F}_\omega$ (with trainable parameters $\omega$).

Given a representative dataset $\mathcal{D} = \{(G^{(n)}, \boldsymbol{S}^{(n)}) : n = 1, \ldots, N\}$ containing $N$ structure-property pairs, the VAE and property predictor are jointly trained as

$$\theta, \phi, \omega \leftarrow \underset{\theta, \phi, \omega}{\arg\min} \ \underbrace{\frac{1}{N}\sum_{n=1}^{N}\left( \left\| \boldsymbol{A}^{(n)} - \boldsymbol{A}^{(n)'} \right\|^2 + \left\| \boldsymbol{x}^{(n)} - \boldsymbol{x}^{(n)'} \right\|^2 \right)}_{\text{reconstruction loss}} + \underbrace{\frac{1}{N}\sum_{n=1}^{N}\left\| \boldsymbol{S}^{(n)} - \mathcal{F}_\omega[\boldsymbol{\mu}^{(n)}] \right\|^2}_{\text{property prediction loss}}$$

$$+ \underbrace{\sum_{n=1}^{N} D_{\text{KL}}\left( \mathcal{N}\left( [\mu_1^{(n)}, \ldots, \mu_d^{(n)}]^\top, \text{diag}\left( [\sigma_1^{(n)2}, \ldots, \sigma_d^{(n)2}]^\top \right) \right) \| \mathcal{N}(\boldsymbol{0}, \boldsymbol{I}) \right)}_{\text{Kullback-Leibler divergence}} . \quad (4)$$

The *reconstruction loss* enforces that the encoded graphs (equivalently, truss lattices) are accurately reconstructed (in terms of both the adjacency matrix and node features) by the decoder. The *property prediction loss* enforces that the property predictor outputs the stiffness of a truss lattice accurately. The Kullback-Leibler divergence (KLD)[52] penalizes the divergence of the probability distribution of the latent space produced by the encoder from the standard Gaussian distribution $\mathcal{N}(\boldsymbol{0}, \boldsymbol{I})$. This allows directly sampling the latent space using a standard Gaussian distribution and decoding truss lattices during the inference stage (as opposed to first encoding an a-priori known lattice into a latent vector and then decoding back during training), which in turn enables the design and discovery of novel trusses beyond the dataset at hand. The KLD loss further simplifies to

$$D_{\text{KL}}\left( \mathcal{N}\left( [\mu_1, \ldots, \mu_d]^\top, \text{diag}\left( [\sigma_1^2, \ldots, \sigma_d^2]^\top \right) \right) \| \mathcal{N}(\boldsymbol{0}, \boldsymbol{I}) \right)$$
$$= \frac{1}{2}\sum_{j=1}^{d}\left[ \sigma_j^2 + \mu_j^2 - 1 - \log\left( \sigma_j^2 \right) \right]. \quad (5)$$

For detailed derivations of the reconstruction and KLD losses, see ref. 52. All details pertaining to the neural network architectures, training protocols, and hyperparameters are presented in Supplementary Table 1.

The generative capability of the VAE enables us to explore novel yet realistic truss structures, whose mechanical properties are immediately available at minimal computational cost through the property predictor $\mathcal{F}_\omega$. With the joint property predictor as a regularizer, the generative modeling framework helps yield a deeper understanding of the latent space, which lacks physical interpretation and hence presents new opportunities for various downstream tasks by modifying the target of the structure-property predictor, e.g., towards the classification of deformation-mechanisms of truss lattices[65], or the prediction of dispersion relations[79,80] and the nonlinear response[50,81]. Furthermore, our framework can be expanded to the simultaneous design of multiple properties by feeding the extracted features to a multi-task property predictor[82–84]. By leveraging the correlations and shared information among different targets, we can effectively guide the design of truss lattices that have various desired properties by integrating the multi-task property predictor into a multi-objective optimization framework.

## Performance of the VAE model

Our first goal is to correctly reconstruct truss structures: any given input lattice is mapped by the encoder into the latent space, from where the decoder reconstructs the truss lattice (Fig. 2a). Defining the topology reconstruction accuracy as the percentage of correctly predicted links reveals that the trained VAE model accurately captures the topological features of trusses with an accuracy score of 99.9% for the adjacency matrix. The correlation plot between the true and reconstructed node positions is presented in Supplementary Fig. 6a. The model shows high quality in the reconstruction of the geometrical features, demonstrated by $R^2 \geq 99.9\%$ across the 3D components ($x$, $y$, $z$) of the node positions. A comparison of representative reconstructed truss structures and the corresponding original structures from the test dataset is shown in Supplementary Fig. 6b.

Next, we assess the performance of the surrogate model for predicting the 3D effective stiffness measures $\boldsymbol{S}$ of trusses on an independent test set. As shown in the correlation plots between the true and predicted stiffness components in Supplementary Fig. 7, the trained model $\mathcal{F}_\omega$ overall achieves an $R^2 \geq 98.2\%$ accuracy across all stiffness components. Altogether, this confirms that our VAE model accurately reconstructs truss structures and predicts their effective stiffness properties.

With the jointly trained property predictor, the latent space is better organized in the sense that structures with similar mechanical properties are expected to cluster in the same region within the latent space (see also Supplementary Note 3.1), which gives important insight into the originally high-dimensional and intractable design space. Moreover, the property predictor works as an additional constraint, enforcing that points in the latent space should decode into valid and realistic truss structures, thus preserving some mechanical property information while reducing the dimensionality. We evaluate the quality and efficiency of the latent space generation by randomly sampling 1000 points from the latent space and using the decoder to reconstruct the corresponding structures. Results show that on average 82.3% (evaluated on 1000 attempts of random sampling) of randomly-selected samples can be successfully decoded into valid (i.e., physically meaningful) truss topologies—we refer to this fraction as the *validity score*. While it is appealing to improve the quality of samples by imposing stronger regularization, such as increasing the weight of the KLD term in Equation (4) (also known as $\beta$−VAE[85]), the fidelity of the reconstruction will degrade due to the inherent trade-off between reconstruction accuracy and random sample quality in VAE models. In this work, we therefore adopt the annealing schedule for the weight of the KL-divergence term[86] (details provided in Supplementary Note 2.1) to dynamically tune the importance given to the regularization and reconstruction losses, thus ensuring the flexibility of the VAE reconstruction.

## Exploration in the latent space of truss lattices

The continuous and low-dimensional latent space with generalization ability is particularly advantageous for the design of new structures by traversing the latent space through simple arithmetic operations of the latent representation $z$. While existing works that relied on a pixel/voxel-based parameterization have shown success in mapping the topology and mechanical properties in a latent space with a similar data-driven design framework[21], they did not consider the impact of the different types of structural features (i.e., of connectivity and node positions) separately. In fact, manipulating the truss connectivity vs. moving nodes will expand the property space differently. For example, changes to the truss topology can have a strong effect on its deformation behavior (stretch- or bending-dominance depends primarily on the connectivity[87]). To this end, we adopt the joint embedding model (details provided in Supplementary Note 2.2) to encode the topological and geometrical features in different dimensions of $z$, while maintaining the total number of latent dimensions constant. A major advantage of this adjustment is that the importance given to each type of information can be adjusted by tuning the corresponding number of latent dimensions without increasing the model complexity. This provides flexibility and allows us to extract information that is only related to the topology or to the node position or to their interactions. As a consequence, traversals through the latent space along different axes give rise to significantly different changes in mechanical properties, which is enabled by the attributed latent embedding approach; i.e., each axis (each component of $z$) stores specific information about the structural pattern transformation. Figure 3 shows an example of moving along three different latent axes (see also Supplementary Information Movies 1–3), which encode the information specifically for connectivity reconstruction, node positions reconstruction, or both—in each case starting from the same truss—which leads to the illustrated changes in structural topology, geometry, or both and the corresponding 3D stiffness (visualized as elastic surfaces).

While previous work has investigated the generation of new structures by moving along a path in latent space[21,78], it is usually intractable to obtain a disentangled representation of the original data space, since the complex correlation between entities is non-trivial to decompose. By contrast, our model uses a systematic latent representation for trusses, which admits human interpretation and where the truss connectivity and node positions can be independently altered. This is useful for many downstream tasks; e.g., for identifying the roles of different geometrical features and their impact on the effective truss performance.

As an illustration, we define an interpolation path between two points in latent space and reconstruct a continuous family of new trusses along the path with the decoder. The high validity score of our latent space ensures that the majority of generated new samples are physically feasible. (Possible issues such as a lack of connectivity or structural instability can be resolved by a light post-processing step.) For example, let us assume that two points in the high-dimensional latent space lie on the surface of a hypersphere rather than on a straight line, so we can interpolate between any two truss structures by applying the spherical linear interpolation[88] (slerp)

$$\text{SLERP}\,(z_1, z_2; \alpha) = \frac{\sin((1-\alpha)\theta)}{\sin\theta} z_1 + \frac{\sin(\alpha\theta)}{\sin\theta} z_2, \quad (6)$$

where $z_1$ and $z_2$ are the vectors of two points in the latent space, $\alpha \in [0, 1]$ is the interpolation parameter, and $z_1 \cdot z_2 = \cos\theta$. A detailed discussion on slerp and its comparison with linear interpolation is provided in Supplementary Note 3.2. Figure 4 presents two examples of interpolating between two truss structures that have significantly different mechanical behavior (e.g., structures with the largest and smallest Young's modulus $E_{11}$, and with the largest and smallest universal anisotropy index $A^U$, which quantifies the degree of structural anisotropy[89] (details are provided in Methods)). Selected structures generated by the decoder along the interpolation path are visualized along with their respective elastic surfaces obtained from FE homogenization (see also Supplementary Movie 4). Our smooth and continuous latent space ensures that, while the start- and end-point structures have opposite extremes of mechanical properties among the dataset, the transition of the structural geometries is smooth. This provides new opportunities for the design of continuous families of truss structures with property grading, which bypasses complex optimization algorithms operating in the high-dimensional, discrete design space. In the second example of Fig. 4 (Supplementary Movie 5), we observed that—along the interpolation path between two points with extreme anisotropy values $A^U$− new structures are generated that have a considerably higher $A^U$-value than all trusses in the training dataset. This is possible, as we interpolate in the latent space instead of the property space. The jointly trained property predictor encourages structures with similar mechanical performance to be located in the same region in latent space. Therefore, sampling in the vicinity of a point in latent space results in a rich family of trusses with similar properties (see also Supplementary Note 3.1). Moreover, new structures generated along a smooth interpolation path, or in the vicinity of the extreme values in the dataset, are expected to exhibit continuous property changes, including unprecedented extreme values.

## Gradient-based optimization in the latent space

The continuous latent space successfully captures the underlying mechanical features of trusses rather than simply memorizing the training data. This enables the use of gradient-based optimization techniques to guide the tailoring of truss lattices to achieve desired properties and, furthermore, to extrapolate beyond the training domain. While the forward mapping from structure to property is straightforward, the inverse design problem is ill-posed due to the one-to-many mapping from the property space to the geometry space (i.e., multiple different truss candidates may lead to the same effective properties). This can be overcome by searching for a potential structure candidate, whose reconstructed stiffness matches the queried stiffness[44]. To generate physically realistic truss structures or to obtain

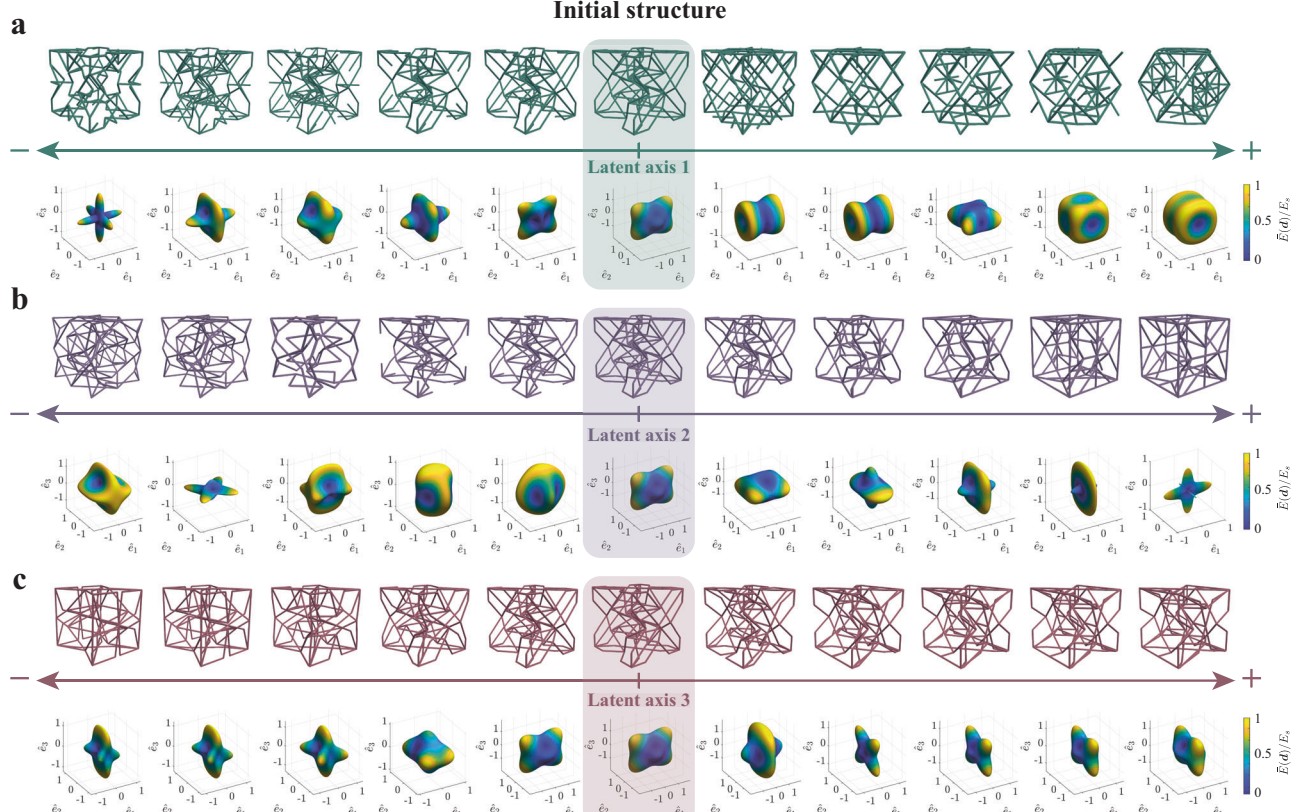

**Fig. 3 | Representative examples of interpolation in the latent space.** Samples are generated by traversals along three different latent axes: **a** taken from the topology-specific, **b** shared topology and geometry, and **c** geometry-specific dimensions of the latent space. Their corresponding 3D elastic surface evolution (obtained by finite element homogenization) is shown along the interpolation path, which indicates the effective directional Young's modulus $E(\boldsymbol{d})$ normalized by the Young's modulus $E_s$ of base material.

the sensitivities of properties with respect to structural features, we leverage automatic differentiation and use a backpropagation algorithm to obtain the gradients through the VAE model and the property predictor. Due to the discrete nature of truss topologies, unconstrained optimization in the latent space can be problematic and may result in invalid structures even with the variational term as a regularization, since there is no explicit constraint on the validity of generated samples when searching the whole latent space. As a remedy, we adopt an indirect approach by first reconstructing truss structures from their latent representation given by the optimizer, and passing them to the encoder to obtain the actual latent variables, which are then forwarded to the property predictor to predict the effective stiffness of generated structures (see Fig. 2). The additional encoding-decoding process ensures that candidate structures proposed by the optimizer are valid.

To demonstrate the inverse design capabilities, we apply our generative modeling framework to design truss structures with extreme mechanical properties. Considering the one-to-many mapping of properties to structures and to have a reference, we first evaluate all structures in the training dataset towards the target property. Based on that data, initial guesses are chosen as the 100 closest matches in terms of the target property. Gradient-based optimizations for each initial guess are performed in parallel, and the best solution is identified by examining the FE-reconstructed properties of the 100 optimal solutions (the property predictor only serves as a computational shortcut to obtain the effective response during optimization). This approach identifies multiple candidate truss structures, which exhibit similar mechanical behavior—allowing for the selection of optimal structures under consideration of additional target attributes such as manufacturability or further

properties of interest (see the detailed discussion in Supplementary Note 3.3).

Figure 5 illustrates three examples of the optimal design of truss lattices towards extreme properties. In all three cases, the trained generative models produce robust designs with properties far outside of the training domain through careful tuning of architectures, thus greatly expanding the limits of the property range. The first example maximizes the directional Young's modulus $E_{22}$, for which Fig. 5a shows the optimization path in the property space. Intermediate truss lattices are visualized to demonstrate the effect of the structural evolution (including the Voigt upper bounds $E_{\text{Voigt}} = E_s \cdot \rho$ and $G_{\text{Voigt}} = G_s \cdot \rho$). The optimization scheme gradually adjusts the beam arrangements along the $\boldsymbol{e}_2$-direction, exceeding the maximum Young's modulus in the training dataset ($E_{22,\text{max}} = 0.068$) by 51.5%. The second example in Fig. 5b shows structures optimized for a maximum auxetic behavior (i.e., for a maximum negative Poisson's ratio $\nu_{21}$ in the $\boldsymbol{e}_1$-$\boldsymbol{e}_2$-plane. The optimization scheme reaches an optimal structure with $\nu_{21} = -2.711$, which is a 42.9% improvement over the most negative Poisson's ratio in the training set ($\nu_{21,\text{min}} = -1.897$). Finally, Fig. 5c illustrates the search for near-pentamode structures[90], i.e., for fluid-like trusses with a high bulk-to-shear modulus ratio—being soft to shear but (close to) incompressible. The gradient optimization scheme here maximizes the ratio of the bulk modulus to the shear modulus. (Since the structures are anisotropic, we use the Voigt average bulk and shear moduli[91], $K_V$ and $G_V$, respectively, for optimization.) Results show how the optimal structure yields a ratio of $K_V/G_V$ that is 28.6% higher than the maximum value contained in the training dataset (which is 14). While this may not be an impressive improvement compared to existing pentamode designs, we stress that—in all three optimization examples—the generative model improved the target

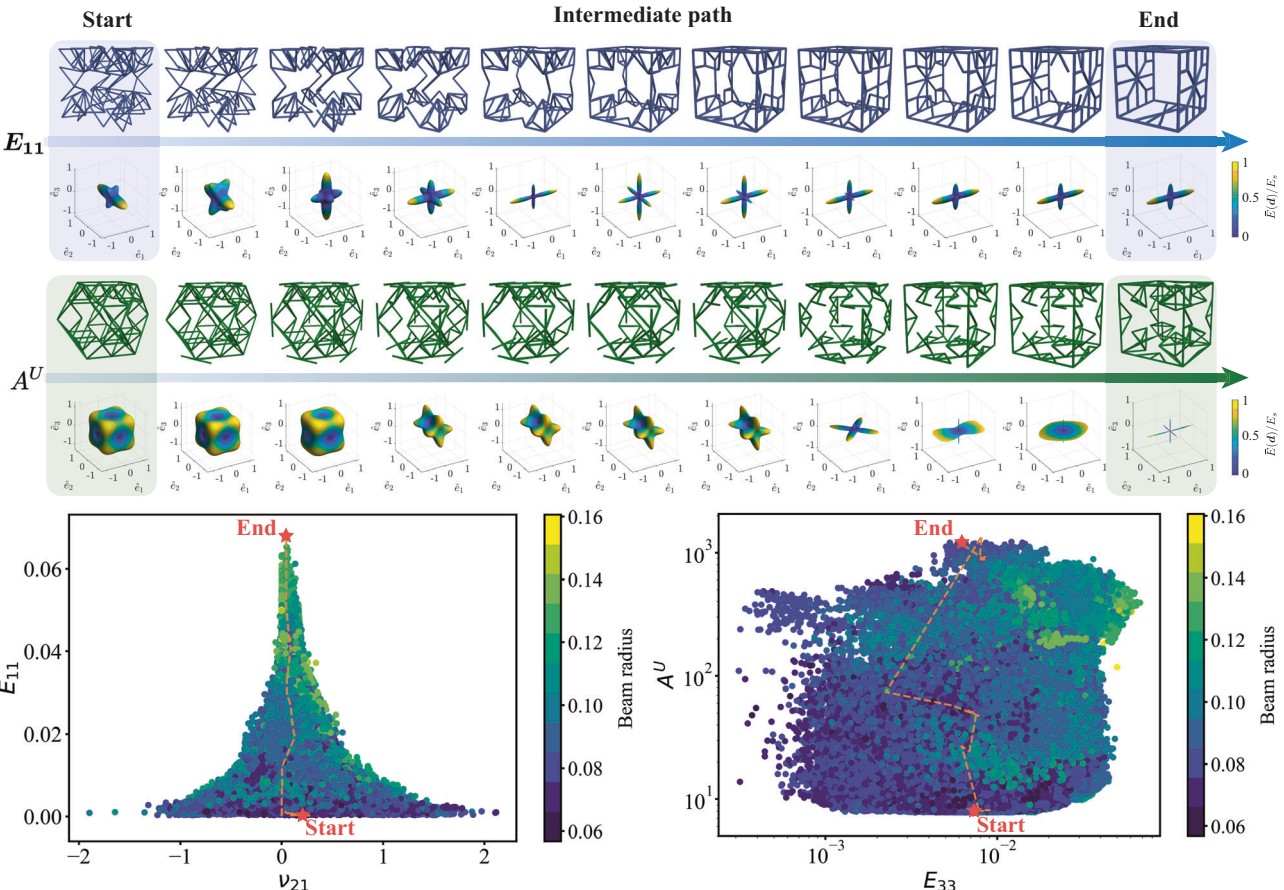

**Fig. 4 | Representative examples of interpolation in the latent space.** Samples are generated by interpolation between two points in latent space, whose corresponding trusses exhibit extreme mechanical properties (in terms of directional Young's modulus $E_{11}$ and the universal anisotropy index $A^U$). Their corresponding 3D elastic surface evolution (obtained by finite element homogenization) is shown along the interpolation path, which indicates the effective directional Young's modulus $E(\boldsymbol{d})$ normalized by the Young's modulus $E_s$ of base material.

properties significantly by only small structural modifications (compare, e.g., the initial and final structures in Fig. 5b and c). Such small changes, which are unlikely to be found by design intuition and experience, demonstrate the complexity of the design and property spaces and highlight the benefits of our approach.

**Design for tailored nonlinear mechanical response**

In addition to target properties in the linear regime, we further verify the efficiency and generalization ability of our framework by the inverse design of *nonlinear* mechanical metamaterials. We consider a subset of the training dataset that contains 383,729 unique structures, striking a balance between computational cost and model performance. To characterize the effective behavior of truss structures, we homogenize the stress–strain response of the truss unit cells with periodic boundary conditions under uniaxial compression subjected to a compressive strain of up to 25% in the z-direction. The established truss database and their corresponding nonlinear stress–strain responses are used to train the generative modeling framework with the objective of enabling the design of novel metamaterials with desired nonlinear responses. To facilitate training of the ML model, we reduce the dimensionality of the learning labels and describe the stress–strain curve by a vector $\boldsymbol{\sigma}_t = [\sigma(0.5\%), \sigma(2.5\%), \ldots, \sigma(24.5\%)]^T$, which contains the compressive stress values at 13 equally spaced strain points along the range of applied compressive strains. Figure 6a shows the comparison between the predicted vs. true stress–strain curves for four representative examples from the test

dataset, exhibiting nonlinear distinct behaviors. The trained property predictor accurately predicts the nonlinear responses for unseen truss structures, achieving an overall normalized root mean square error (NRMSE) of 4.5%, which confirms that the property predictor provides an effective estimate of the nonlinear responses of diverse trusses.

Next, we demonstrate the inverse design capabilities of the model by applying our generative modeling framework to design truss structures matching a given target stress–strain response. Figure 6b shows two benchmark examples of different stress–strain response targets. First, we select a design target that exceeds the stiffest response in the considered training dataset by 30%. As shown in Fig. 6b (i), the optimal truss structure closely matches the target response with an NRMSE of 3.8%, which showcases the capability of the framework to design truss structures that exhibit specific desired responses, even beyond the range of observed behaviors in the training dataset. Second, we consider a target stress–strain curve displaying pronounced softening behavior, with a minimum NRMSE between the top pick within the training dataset and the target of 9.1%, as illustrated in Fig. 6b (ii). The generated optimal design significantly outperforms the best match in the training dataset for the considered target (with an NRMSE of 3.0%). This demonstrates that the constructed continuous latent space effectively captures the essential features and (some of them) underlying physics of periodic trusses, which enables the inverse design of novel truss designs that closely match unseen responses in both the linear and nonlinear regimes, clearly surpassing the limits of the training dataset.

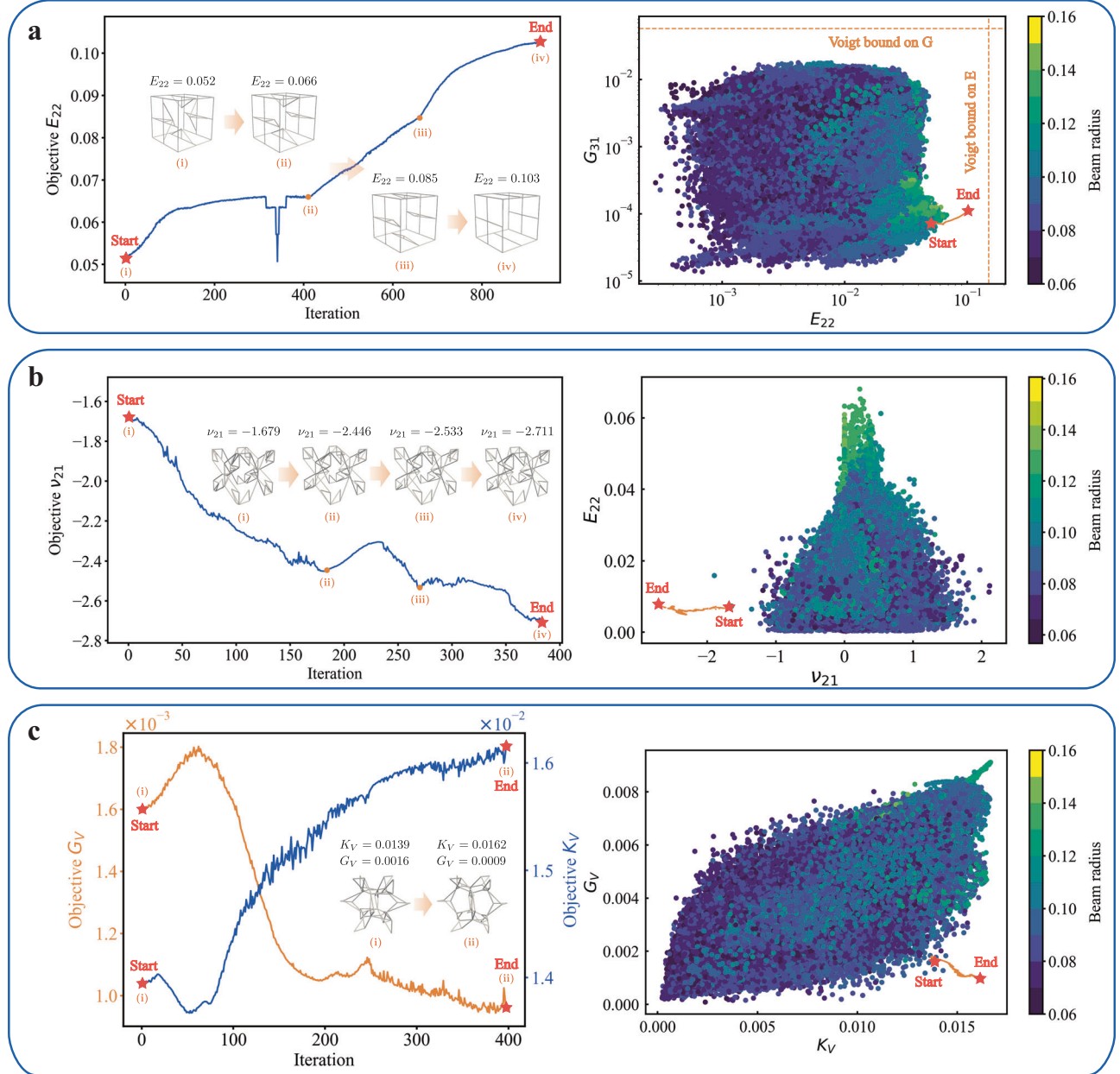

**Fig. 5 | Inverse-designed truss metamaterials based on gradient optimization.**
**a** Maximizing Young's modulus $E_{22}$, **b** minimizing Poisson's ratio $\nu_{21}$, and
**c** maximizing the bulk-to-shear modulus ratio $K_V/G_V$ of truss lattices. Each example
shows the property evolution vs. the number of optimization iteration steps,
including a few selected structures at the indicated points, as well as the property
path compared to the training dataset in the relevant property spaces (each dot
represents a truss in the training data, which is color-coded by the (dimensionless)
radius of the beams with circular cross-section). Source data are provided as a
Source Data file.

## Discussion

The presented generative modeling framework constructs a continuous,
low-dimensional latent space of truss metamaterials. By analogy with
molecules, we leverage the graph representation to interpret periodic
trusses as graphs, thus providing an efficient, consistent, and general
parameterization, which covers a wide range of truss structures and a
tremendous space of anisotropic mechanical properties. Encoding the
information related to the truss connectivity, the node positions, and
their shared information in different dimensions of the latent repre-
sentation enables a human interpretation of the otherwise intractable
latent space. It also provides flexibility and tunability in manipulating
structural features of truss lattices to achieve optimal properties. A
major advantage of the unified and continuous latent representation is
that novel truss structures can be conveniently generated by simple

operations in the latent space, including sampling in the vicinity of
known data points, traversing along the latent axes, and interpolating
between two points. While classical VAEs often suffer from the issue of
opacity and a lack of a physically meaningful representation, the pro-
posed framework tackles this challenge by a jointly trained neural net-
work to predict the truss properties from the latent space—thus allowing
us to creatively navigate the latent space and to extrapolate with
gradient-based optimization techniques to unseen, extreme properties
outside the original training domain. This admits identifying optimal
lightweight truss lattices with target combinations of, e.g., the elastic
constants in 3D and nonlinear stress–strain responses. The proposed
design framework admits extension to other properties of truss
metamaterials[45,92,93] by modifying the property predictor as well as to
other types of metamaterials[37,44,94–97] by modifying the design

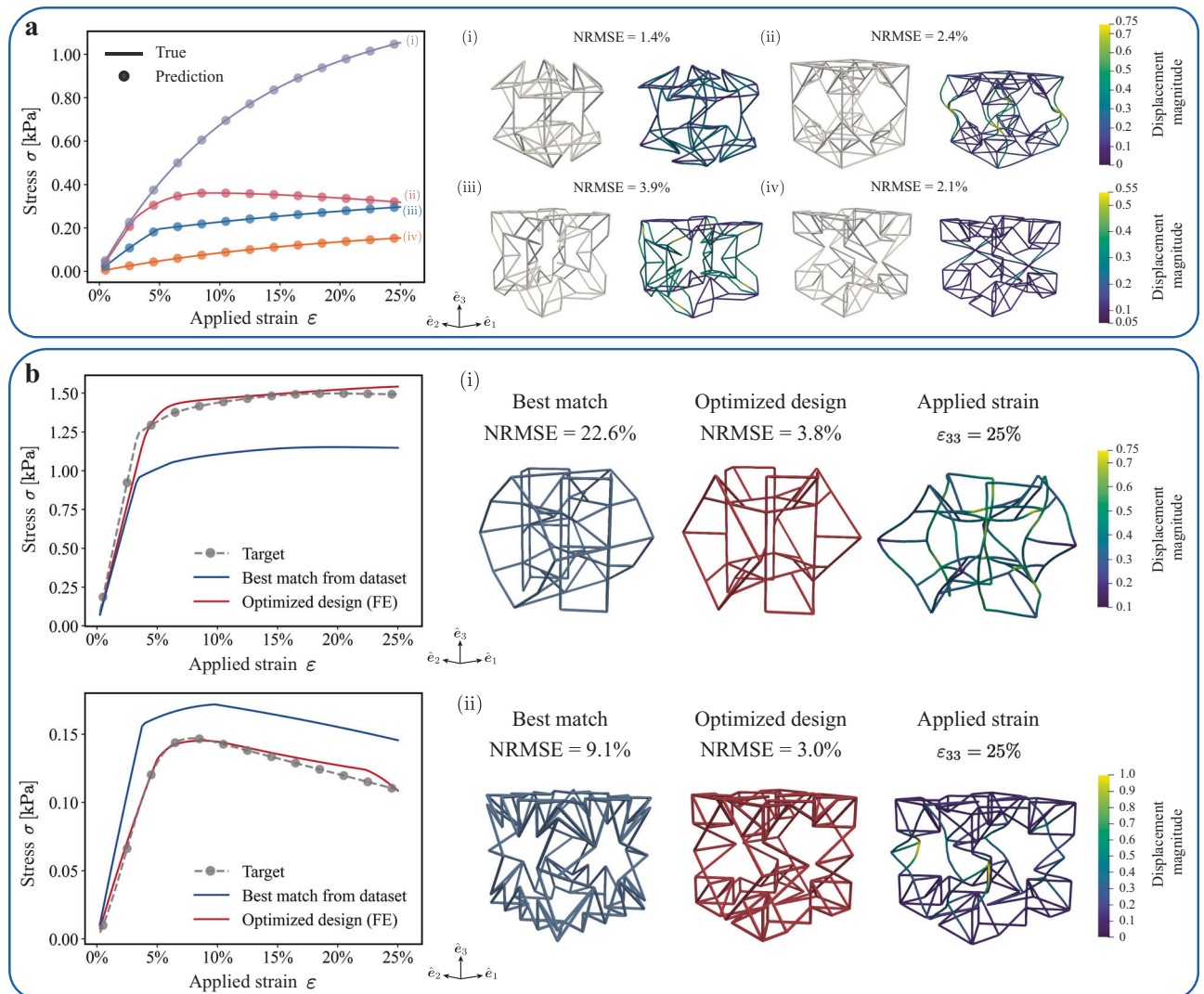

**Fig. 6 | Generative design of truss metamaterials with target nonlinear response. a** Comparison between the stress–strain curves predicted by the property predictor (solid dots) and the ground truth (solid lines) for four representative truss structures. Each of the four examples shows the corresponding truss unit cell and its (dimensionless) displacement magnitude under an applied compressive vertical strain $\varepsilon_{33} = 25\%$. **b** Inverse-designed truss structures obtained from gradient optimization with target nonlinear responses. Each example shows the comparison of responses between the best match from the training dataset and the optimized solution as well as the corresponding truss unit cells. All shown true stress–strain curves were obtained by finite element homogenization.

parameterization. This highlights the potential of our framework to analyze and optimize a broad range of metamaterials. The physical interpretability and extrapolation ability open up new avenues for the discovery of new metamaterials and lend inspiration for designing cellular structures with tailored properties by tuning the architectural features.

## Methods
### Data generation
Supplementary Fig. 1 shows the details of the generation of a diverse truss lattice dataset. We define the truss graph in the octant within a domain $\Omega_0 = [0,1]^3 \subset \mathbb{R}^3$, which is populated into truss structures in the domain $\Omega = [-1,1]^3 \subset \mathbb{R}^3$ through reflections about the three mutually orthogonal symmetry planes. Starting from the five elementary truss lattices shown in Step 1 of Supplementary Fig. 1, new structures are created by randomly perturbing both the node positions and connectivities for several iterations. Supplementary Fig. 2 shows the node positions and connectivities of the five considered elementary truss structures. Node positions are altered by offsets defined in the natural coordinate system[66] and sampled from the uniform

distribution $\lambda \sim \mathcal{U}(-0.5, 0.5)$. Based on the initial truss structures, new connectivities may be introduced by removing available nodes or inserting new nodes with the following constraints: (1) the established structure is a single connected component; (2) all beams are shorter than $r_{max} = \sqrt{3}/2$ (times the unit side length of the unit cell); the maximum permissible length of a beam connection $r_{max}$ is chosen according to the longest connection in the initial five elementary trusses; (3) no dangling connections exist within the structure (every node has at least two connected beams). Each geometry is perturbed for 10 iterations to generate a library that contains a wide range of truss structures with several unique topologies. From the established set, we randomly sample two lattices with repetitions allowed, which are then superimposed according to their matching nodes to yield a more diverse dataset. The full dataset considered for training the generative models contains 965, 736 lattices and their homogenized effective stiffness properties.

### Computational homogenization
The effective stiffness tensor $\mathbb{C}$ of all truss structures in our library is computed by FE homogenization with periodic boundary conditions[75],

using on an in-house C++ FE code (available at http://ae108.ethz.ch). Each strut in the truss unit cell is modeled as a linear elastic Timoshenko beam with a circular cross-section. The strut radius of each unit cell is scaled to maintain a constant relative density of $\rho = 0.15$. We visualize the 3D anisotropic stiffness of truss lattices as elastic surfaces, which indicate the effective directional Young's modulus $E(\boldsymbol{d})$ for all directions $\boldsymbol{d} \in S^2$ as

$$E(\boldsymbol{d}) = \left( \sum_{i,j,k,l=1}^{3} \mathbb{C}_{ijkl}^{-1} d_i d_j d_k d_l \right)^{-1}. \tag{7}$$

In Figs. 3 and 4, we presented representative samples of novel truss lattices generated by interpolating between known structures that exhibit extreme universal anisotropy values $A^{U[89]}$. $A^U$ can be interpreted as a generalization of the Zener index[98], which applies to structures with cubic symmetry and can be expressed as

$$A^U = 5\frac{G_V}{G_R} + \frac{K_V}{K_R} - 6, \tag{8}$$

where $G_V$ and $K_V$ are the Voigt estimates for, respectively, the shear and bulk moduli[91], and $G_R$ and $K_R$ are the Reuss estimates for, respectively, the shear and bulk moduli[99]. Specifically, Reuss proposed the following relations for the bulk modulus $K$ and shear modulus $G$ in terms of the compliance components $\mathbb{S}_{ijkl}$:

$$K_R^{-1} = (\mathbb{S}_{1111} + \mathbb{S}_{2222} + \mathbb{S}_{3333}) + 2(\mathbb{S}_{1122} + \mathbb{S}_{1133} + \mathbb{S}_{2233}), \tag{9}$$

$$15G_R^{-1} = 4(\mathbb{S}_{1111} + \mathbb{S}_{2222} + \mathbb{S}_{3333}) - 4(\mathbb{S}_{1122} + \mathbb{S}_{1133} + \mathbb{S}_{2233}) + 3(\mathbb{S}_{4444} + \mathbb{S}_{5555} + \mathbb{S}_{6666}). \tag{10}$$

Analogously, we adopt the Voigt average bulk and shear moduli, calculated from the anisotropic stiffness components $\mathbb{C}_{ijkl}$ as, respectively,

$$9K_V = (\mathbb{C}_{1111} + \mathbb{C}_{2222} + \mathbb{C}_{3333}) + 2(\mathbb{C}_{1122} + \mathbb{C}_{1133} + \mathbb{C}_{2233}), \tag{11}$$

$$15G_V = (\mathbb{C}_{1111} + \mathbb{C}_{2222} + \mathbb{C}_{3333}) - (\mathbb{C}_{1122} + \mathbb{C}_{1133} + \mathbb{C}_{2233}) + 3(\mathbb{C}_{4444} + \mathbb{C}_{5555} + \mathbb{C}_{6666}). \tag{12}$$

## ML framework
Details of the optimized dimensions and hyperparameters (e.g., the number of hidden layers and nodes in each layer, activation functions, learning rates, etc) of the VAE model and the property predictor are provided in Supplementary Table 1. Thresholding is applied at the end of the connectivity decoder by a sigmoid function to achieve a binary connectivity matrix. 1% of the generated dataset is used for the tuning and optimization of hyperparameters. We used the PyTorch[100] package throughout the implementation of the proposed generative learning framework and leveraged its automatic differentiation engine, autograd, to automatically obtain the gradients of the homogenized effective properties with respect to the structural and geometrical features towards the optimization and inverse design of truss lattices. To remedy the issue of vanishing KL-divergence term[101], we schedule the weight of the KL-divergence term $\beta$ via the cyclical schedule[86] shown in Supplementary Fig. 3. The training process is split into several cycles, and in each cycle $\beta$ is gradually increased from 0 to 1, using a linear function after 50 epochs. For a detailed performance comparison of various annealing schemes for $\beta$, see refs. 86,101,102.

Details of the data generation (Section 1), the ML protocols (Section 2.1), the implementation of the overlapping embedding

model (Section 2.2), the NN model performance (Section 2.3), exploration in the latent space including sampling (Section 3.1) and interpolation (Section 3.2), details on the gradient-based optimization in the latent space (Section 3.3), details on the inverse design of truss metamaterials with target nonlinear responses (Section 3.4), and the computational efficiency estimates (Section 4) are summarized in the Supplementary Information.

## Data availability
The training data including truss structures and their effective homogenized properties generated in this study have been deposited in the ETHZ Research Collection[103]. Source data are provided with this paper.

## Code availability
The code used to train the generative modeling framework and obtain inverse designs of truss structures has been uploaded to Github[104]. The FE code used for homogenization in this study is available in the ae108 library[105].

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

## Acknowledgements
This research received financial support from Adidas as well as from ETH Zurich through the ETH+ grant SynMatLab. K.K. acknowledges the support from a Marie-Sklodowska Curie Postdoctoral Fellowship under Grant Agreement No. 101024077. The authors gratefully acknowledge the support from Adidas and the discussions with Dr. Ladan Salari-Sharif and Derek Luther.

## Author contributions
**L.Z.**: Methodology, Software, Validation, Data Curation, Visualization, Writing—Original Draft; **K.K.**: Software, Data Curation; **S.K.**: Conceptualization, Methodology, Writing—Review & Editing, Supervision; **D.M.K.**: Conceptualization, Methodology, Writing—Review & Editing, Supervision.

## Competing interests
The authors declare no competing interests.
