## [Peer Review File · Nature Communications]

REVIEWER COMMENTS

Reviewer #1 (Remarks to the Author):

In the manuscript “Unifying the design space of truss metamaterials by generative modeling”, a deep learning framework based on a graph representation and variational autoencoder neural network is introduced to model the design space of 3D truss lattice metamaterials and efficiently optimize them. The graph structure combines a discrete representation of the topology of the lattices with continuous node features, here the variations of positions. The mechanical properties of the lattices in terms of the coefficients of the linear elastic constitutive tensor are not determined directly in terms of the graph features but based on the latent description that is determined through the autoencoder, which enables efficient design optimization.

The approach taken in this work is sound and well executed, and the results presented are convincing and impressive. However, a significant novelty over existing works that would justify publication in NatComms is not entirely clear to me.

Major comments:

In the introduction, the differences and novelties over existing works should be clarified, especially the ones that also employ graph NNs for modeling lattice structures, such as refs [47,59,65], <https://doi.org/10.1016/j.matdes.2022.111175>, “E. Ross & D. Hambleton: Using Graph Neural Networks to Approximate Mechanical Response on 3D Lattice Structures, Advances in Architectural Geometry 2020”.

Important claims of the work are that it is beneficial to use the latent space, that the latent space offers interpretability, and that “any two lattices with similar topological and geometric features are located close to each other”. However, it is not mentioned how the dimension(s) of the latent space are determined (according to the supplementary, it is 40+40 or 48) and how sensitive the results would be to the size of the latent space. Given that is still relatively large (40 vs. 278+27), it is also not clear what the advantage over direct use of the direct inputs would be. Furthermore, the interpretability of the latent space variables is also not apparent to me and changes in the elastic properties seem sometimes rather abrupt in Fig. 3b and the movies S1 & S2. How is it ensured and demonstrated that “any two lattices with similar topological and geometric features are located close to each other” and that “similar mechanical properties are expected to cluster in the same region within the latent space”? What makes the latent representation “systematic” and how does it “admit human interpretation”?

The definition of the “maximal” topology is rather arbitrary, some existing works consider less vertices, but maybe even more fine-grained topologies would enable even more extreme properties? Furthermore, here lattices with constant strut diameters are considered, but couldn’t varying or graded struts also be considered, see e.g., <https://doi.org/10.1002/nme.6869>, which would directly lead to a continuous topology parameterization?

Thus the claim of the present graph representation being “generalizable and unified” is over-emphasized in my opinion.

Minor comments:

In Fig. 1a+b, it would be helpful to consistently color the nodes as vertex, edge, face & body nodes.

On page 3, the lattices and elastic tensors are referred to as generally “anisotropic”, but I believe it would be better to directly mention the symmetry, which should be rhombic in my opinion.

The correlation plots for assessing the performance of the VAE look generally good, but Fig. S4a-y shows a few extreme outliers. What might be the reason and how could this be improved? What are the consequences of such outliers?

Furthermore, in Fig. S5 especially the shear components are not approximated so well - what could be the reason? Maybe it would help to predict the Cholesky decomposition instead of the components of the elasticity tensor to preserve positive definiteness? See <https://doi.org/10.1016/j.jcp.2020.110072>, <http://arxiv.org/abs/2203.13938>

Since the decoding accuracy is only 82%, what should one do when an optimization result cannot be decoded successfully?

Page 9, line 2: A number is missing after “Section”

Reviewer #2 (Remarks to the Author):

The manuscript 'Unifying the design space of truss metamaterials by generative modeling' is well written and novel piece of work. This paper provides a graph-based deep generative network that combines a variational autoencoder and a structure-property predictor. The discrete design space for trusses is reduced to continuous latent space, thus enabling optimization in a continuous space to obtain truss structures with desired properties.

The reviewer recommends publication of the article after the following queries are addressed satisfactorily:

- 1) It is quite innovative to decompose the latent space dimensions into topology, geometry, and their shared dimensions. However, it would be nice to spell out the procedure of the decomposition in relatively more details. Also, it is not entirely clear how the dimensions of different subspaces, such as topology-, geometry-specific spaces, are determined.
- 2) The framework seems to be general enough to be applicable to different kinds of metamaterials, such as plate- and shell-lattices. However, the method demonstrated here for truss-lattices. Can the framework be extended to other kind of lattices, for example shell-lattices? If not, could the author suggest the required modifications that are necessary to the framework?
- 3) The author focused only the elastic behavior of the truss metamaterials. However, it appears that just by modifying the structure-property predictor, it might be possible to optimize for the other properties, such as toughness, dispersion relations, as well. If this is correct, could the authors provide a brief discussion about the plausibility of extending the work for simultaneously optimizing two or more properties?
- 4) As have been mentioned by the authors, there are multiple truss structures that have similar properties. However, in Fig. 4, which demonstrates the optimal truss structures for desired properties, shows only one such optimal lattice structure. It would be nice to provide the details, possibly in the supplementary, as to how the initial structures are chosen, their properties at different optimization iterations and some of the optimal lattice structures that have similar properties.
- 5) The authors leverage the graph representation of the truss based lattices. Have the authors considered permutation and equivariance for these graph representations i.e. if a lattice is rotated in the Euclidean space, will the graph interpret the two lattices as identical or different?

Minor comments:

- 1) The authors have mentioned that spherical interpolations in Eq. (6). However, it is not clear whether linear or spherical interpolations are used in interpolations in the latent space, for example in Fig. 3. If spherical interpolations are used, it is not entirely clear why such interpolations, instead of linear interpolations, are necessary.

2) On Page 9, the sentence “To this end, we adopt the joint embedding model presented in Section to encode the topological and geometrical features in different dimensions of z , while maintaining the total number of latent dimensions constant. “seems to have a missing section reference.

3) Are the elastic surfaces in Fig. 3 are predictions or ground truth from numerical calculations?

4) On Page 4, the authors mention “(e.g., the representative examples shown in Figure1d reach effective Young’s moduli of ca. 36% higher than that of a simple cubic unit cell in the principle direction at the same density)”. Can the authors clarify that this is possibly true for just one direction.

5) On Page 7, the authors mention “ Results show that on average82.3%(evaluated on 1,000 attempts of random sampling) of randomly-selected samples can be successfully decoded into valid (i.e., physically meaningful) truss topologies – we refer to this fraction as the validity score” What about the accuracy of stiffness prediction on those?

Reviewer #3 (Remarks to the Author):

In summary, this manuscript is well-written and could be accepted after several minor revisions. Here are the comments:

1. In the second part of Results, it is mentioned that the proposed deep learning method can avoid the expensive computation cost of the FE homogenization computation. However, neural network surrogate model is still based on the finite element simulation. How to define this cost for the proposed method?

2. The effective stiffness-related 3D coordinate labels in Figures 2 and 3 are not clear. The involved figures should be further improved.

3. What are the criteria for classifying the five elementary truss lattices in Section 1 of Supplementary Information? Some necessary explanation should be given.

4. The meaning of “ r_{max} ” in the conditions for adding new nodes mentioned in the Section 1 of Supplementary Information is also need to be explained.

Reviewer #4 (Remarks to the Author):

The authors have described a very good and interesting topic. They have proposed a graph-based deep learning generative framework, and present a VAE model containing an encoder and a decoder to optimize by neural network surrogate model without costly FE homogenization computation. Thereafter, the authors have assessed the accuracy and flexibility of the VAE model. Finally, they study the possibility of using VAE model to design new configurations based on the latent space and the gradient-based optimization in the latent space.

Overall, this work is of significance to the related fields. However, to allow the submitted manuscript to be accepted by such a well-renowned journal, some comments should be further concerned by the authors:

- a). To allow the work to be reproduced, some important details (e.g., axial stiffness and length of the members) should be provided.
- b). Both graph theory (with nodal positions considered, i.e., geometric-graph-theoretic) and ML are adopted in this study. In fact, it would be much better if the graph theory is combined with symmetry (or group theory).
- c). This reviewer is very interested in whether the RVW has been selected under relatively fair conditions (such as the same density, same volume, same mass) during the ML process.
- d). As far as engineering applications are concerned, the authors should consider potential difficulty of the connectivity of the joints, and potential local buckling of the slender members under compression.

Reviewer #5 (Remarks to the Author):

The paper presents a graph-based deep learning generative framework to explore an extremely large design space for truss lattices. The method combines existing generative machine learning models, a variational autoencoder and an adversarial network, which have been already successfully used to design metamaterials. Graph-based learning has been demonstrated successful across several domains including polycrystalline materials due to the capacity to model interrelations in irregular domains. The graphical representation used by the authors in this paper encodes the geometric descriptors defining the truss lattice, such as strut coordinates, volume fraction, nodal connectivity, and strut cross-section, into a data structure that can in turn be used to optimize truss lattices with unusual mechanical properties on target.

While I certainly find the contents of this work scientifically solid, I also find striking similarities to ref 47, a recent paper published by some of the authors. The choice of the fundamental unit cell with cubic symmetry, the machine learning scheme, the inverse design approach, the computation of the

homogenization elastic properties, and the formulation of the optimization problem to tune the elastic properties. The difference seems to lie mainly on the choice of the generative model for the truss descriptors. In addition, the generative frameworks does rely on existing models which have been already successfully applied in other domains.

Given the above, I don't see sufficient ground to warrant publication in this journal, although the technical work is sound.

Unifying the design space of truss metamaterials by generative modeling

Li Zheng, Konstantinos Karapiperis, Siddhant Kumar, and Dennis M. Kochmann

We are grateful to all five reviewers for their constructive and insightful comments as well as the editor for allowing us to respond to those. This revision has led to a significantly improved version of our submission, including new simulations, extended methods, and especially a new extension of our framework to nonlinear material behavior. In the following, we respond to each point raised by the reviewers. All changes made to the manuscript and the supplementary material are highlighted in red in the enclosed revised documents.

Kindly note that we added Dr. Konstantinos Karapiperis (ETH Zurich) to the list of authors, as he performed the new simulations of the nonlinear mechanical response of structures, which we report in the revised manuscript.

Reviewer 1

In the manuscript “Unifying the design space of truss metamaterials by generative modeling”, a deep learning framework based on a graph representation and variational autoencoder neural network is introduced to model the design space of 3D truss lattice metamaterials and efficiently optimize them. The graph structure combines a discrete representation of the topology of the lattices with continuous node features, here the variations of positions. The mechanical properties of the lattices in terms of the coefficients of the linear elastic constitutive tensor are not determined directly in terms of the graph features but based on the latent description that is determined through the autoencoder, which enables efficient design optimization.

The approach taken in this work is sound and well executed, and the results presented are convincing and impressive. However, a significant novelty over existing works that would justify publication in NatComms is not entirely clear to me.

Response 1.0: We thank the reviewer for their kind feedback and will gladly respond to the points raised by the reviewer and outline the novelty of our work.

Major comments:

Reviewer 1.1: *In the introduction, the differences and novelties over existing works should be clarified, especially the ones that also employ graph NNs for modeling lattice structures, such as refs [47,59,65], <https://doi.org/10.1016/j.matdes.2022.111175>, “E. Ross & D. Hambleton: Using Graph Neural Networks to Approximate Mechanical Response on 3D Lattice Structures, Advances in Architectural Geometry 2020”.*

Response 1.1: In the revised manuscript, we have expanded the explanation of the (significant) advantages of our approach over existing graph-based methods for modeling lattice structures. Allow us to elaborate here without the strict length limitation of the manuscript.

Graph learning has recently shown promising results in mapping from geometries to the property space of lattice architected materials, owing to the unique non-Euclidean structure of graphs and their inherent resemblance to truss-based lattices. Once trained, graph NN surrogate models can bypass costly simulations and provide real-time prediction on various properties of interest, e.g., homogenized elastic properties¹ and thermal properties², dominant deformation mechanism³, etc. from lattice geometries, thus offering a significant advantage in accelerating the design application of lattice materials. In contrast to the above supervised graph-based models focusing on the forward prediction of property from structures (including the work of Ross et al. pointed out by the reviewer), the goal of the *graph generative modeling* framework proposed in this work is to construct a unified, *continuous* latent representation of a vast and discrete truss design space and the *inverse design* for targeted mechanical properties. Importantly, our framework is not limited to linear properties but – as we demonstrate in our revised manuscript – can be well generalized to design trusses with target *nonlinear* responses (see Figure 5 in the revised article, which demonstrates the generative design of architectures for a given nonlinear stress-strain target curve; also reproduced here in Figure 1). By learning from existing data, the model captures the underlying data distribution, which allows us to directly generate diverse and previously unseen lattice structures by interpolation and extrapolation, which brings promising opportunities in optimizing novel graph configurations and pushing the boundaries of the design space (as also demonstrated in other fields such as drug discovery⁴ and protein structure design⁵).

Figure 1. Generative design of truss metamaterials with target nonlinear response: (a) Comparison between the stress-strain curves predicted by the property predictor (solid dots) and the ground truth (solid lines) for four representative truss structures. Each example shows the corresponding truss unit cell and its displacement magnitude under an applied strain of $\epsilon_{33} = 25\%$. (b) Inverse-designed truss unit cells obtained from gradient optimization with target nonlinear responses. Each example shows the comparison of responses between the best match from the training dataset and the optimized solution, as well as the corresponding truss in its deformed configuration. All shown true stress-strain curves were obtained by FE homogenization.

Furthermore, the graph-based parameterization of truss lattices proposed in this work promises an intriguingly large design space (larger than all prior approaches). Inspired by the decomposition approach in refs. ^{1,6}, which primarily focused on isotropic truss unit cells, we have developed the methodology to both model and design a diverse database of *anisotropic* (orthotropic) truss lattices with the resulting property space spanning a wide range of mechanical properties, as illustrated in Figure 1 of the manuscript. Compared to ref. ¹, our design space considers significantly more nodes (and variable numbers of nodes), assumes less symmetries, and the dataset used covers a tremendous space of truss topologies (compared to a single topology used there). Compared to existing works that rely on a catalog of representative unit cells selected in a heuristic manner ⁷⁻⁹, the graph representation allows great flexibility in exploiting the broad range of available truss topologies as any pattern of truss connectivity can be naturally described by graphs with edges denoting struts and nodes denoting their intersections. Moreover, the graph-based parameterization can generalize well to a wider design space by introducing additional design parameters as graph

features (see Response 1.5), which enables us to enrich the design space by tuning geometry features (e.g., individual strut diameter¹⁰) or base material properties (e.g., relative density¹¹). The graph-based approach thus offers a powerful framework for exploring and optimizing truss lattice designs, providing an avenue for innovative solutions in a range of applications.

To summarize, particularly in contrast to the state-of-the-art work of Ross et al.¹ pointed out by the reviewer, we tackle a much more challenging problem of (i) generative modeling and inverse design (as opposed to simply forward property prediction) for (ii) targeted anisotropic mechanical properties (as opposed to isotropic lattices) and (iii) the design for tailored *nonlinear* stress-strain responses.

Reviewer 1.2: *Important claims of the work are that it is beneficial to use the latent space, that the latent space offers interpretability, and that “any two lattices with similar topological and geometric features are located close to each other”. However, it is not mentioned how the dimension(s) of the latent space are determined (according to the supplementary, it is 40+40 or 48) and how sensitive the results would be to the size of the latent space. Given that is still relatively large (40 vs. 278+27), it is also not clear what the advantage over direct use of the direct inputs would be.*

Response 1.2: We appreciate the reviewer’s comment and have added more details regarding the dimensionality of the latent space in Section 3.2 of the revised Supplementary Information. The choice of the dimensionality of the latent space is determined empirically, balancing the trade-off between the quality of the reconstruction accuracy and the computational cost based on our experimental findings. Figure 2 below schematically illustrates the structure of the overlapping embedding model. Adjacency matrix reconstruction relies on d_A dimensions, node position reconstruction relies on d_x dimensions, and d_{Ax} overlapping dimensions are shared between them. We performed a systematic study to investigate the performance of the model with different latent space dimensions, as shown in Table 1. We observe that increasing the shared dimensions allocated to both beyond a certain point did not significantly improve the model’s performance. Therefore, we selected the latent dimension $d_A = 8$, $d_{Ax} = 32$, $d_x = 8$, which strikes a reasonable balance between capturing essential features of truss structures and maintaining computational efficiency.

Figure 2. Diagram of the overlapping embedding model. $d_A - d_{Ax}$ is the number of latent dimensions allocated exclusively to topology; $d_x - d_{Ax}$ is the number of dimensions allocated exclusively to geometry; d_{Ax} is the number of dimensions allocated to both.

R^2 -scores	A	(x, y, z)	C_{1111}	C_{1122}	C_{1133}	C_{2222}	C_{2233}	C_{3333}	C_{2323}	C_{3131}	C_{1212}
$d_A = 20, d_{Ax} = 8, d_x = 20$	0.990	0.995	0.985	0.957	0.974	0.976	0.967	0.982	0.970	0.974	0.968
$d_A = 16, d_{Ax} = 16, d_x = 16$	0.990	0.996	0.972	0.969	0.962	0.989	0.978	0.932	0.991	0.971	0.955
$d_A = 8, d_{Ax} = 32, d_x = 8$	0.999	0.999	0.995	0.987	0.989	0.995	0.990	0.995	0.983	0.983	0.982
$d_A = 4, d_{Ax} = 40, d_x = 4$	0.989	0.994	0.965	0.990	0.958	0.977	0.983	0.954	0.969	0.958	0.973

Table 1. Comparison of the node positions reconstruction accuracy of the VAE model and the prediction accuracy of the property predictor using different latent dimension d_A, d_{Ax}, d_x . The coefficient of determination R^2 -scores are used as the primary metric.

Regarding the advantage of using the proposed latent representation over the direct use of the input features, the

latent space provides several benefits. First, the generative model encodes a rich truss structure database (containing millions of structures) into a *compressed latent space*, which allows for easier manipulation and interpretation. It effectively reduces the dimensionality of the data, allowing for more efficient processing and analysis. By capturing the essential features in a lower-dimensional space, computational complexity can be significantly reduced, enabling faster computations and improved scalability for tasks like optimization and inverse design.

Second, the latent space provides a uniform and *continuous* representation of the discrete design space. The original input is discrete and of combinatorial nature (one may add or remove nodes, change the connectivity between nodes, change the number of connections, etc.). By contrast, the continuity of the latent spaces enables *smooth interpolation and extrapolation* between different designs. For instance, novel structures can be generated simply by sampling in the latent space and passing the new latent vectors to the decoder without the need to perform iterative computations. As the original input lacks inherent continuity, it is significantly more challenging to navigate the resultant design space or perform optimization.

Third, the latent space obtained through a Variational Autoencoder (VAE) captures the underlying structure and relationships within the data. The jointly trained property predictor drives the VAE model to encode truss structures in such a way that the latent representations are organized according to their mechanical properties for better regression performance (see the detailed discussion in Response 1.3). This can be highly advantageous for downstream optimization tasks. By leveraging the encoded representations, it becomes easier to explore different design possibilities and search for optimal solutions efficiently.

Fourth, the truss networks studied here have the inherent challenge that small changes to the design can lead to considerable changes in the effective properties. For example, consider a straight strut that is replaced by two struts under an angle close to (but not exactly) 180° . While the straight strut offers maximum stiffness along its length, the introduction of the small deviation by an offset middle node can significantly reduce the stiffness by promoting bending deformation. Considering a complex truss with many struts, such small changes to the design input can yield large changes in the property output. By contrast, the latent space representation obtained from the VAE aims for nearby points in latent space having similar properties (see also Response 1.3 below). Therefore, the latent space enables us to identify (following optimization) not only a single structure but many (potentially dissimilar) structures with similar properties in the vicinity of the optimal design in latent space (which would not be the case when using the original input).

In summary, although the dimensionality of the latent space is not by far lower than the original input, the uniform and continuous latent space representation offers key benefits.

Reviewer 1.3: *Furthermore, the interpretability of the latent space variables is also not apparent to me and changes in the elastic properties seem sometimes rather abrupt in Fig. 3b and the movies S1 & S2. How is it ensured and demonstrated that “any two lattices with similar topological and geometric features are located close to each other” and that “similar mechanical properties are expected to cluster in the same region within the latent space”? What makes the latent representation “systematic” and how does it “admit human interpretation”?*

Response 1.3: We thank the reviewer for raising this point and have added a comment in Section 3.4.1 of the revised Supplementary Information. To illustrate that the VAE model tends to encode lattices with similar topologies and mechanical properties in the same region of the latent space, we randomly picked a starting point \mathbf{z}_0 in the latent space and sampled points in its vicinity by adding Gaussian noise to the point according to

$$\mathbf{z} = \mathbf{z}_0 + \beta \cdot \boldsymbol{\varepsilon}, \quad \text{with } \boldsymbol{\varepsilon} \sim \mathcal{N}(\mathbf{0}, \mathbf{I}), \quad (1)$$

where $\beta \in [0, 1]$ is a scaling factor that determines the range of sampling. Figure 3c shows some representative examples of structures obtained by decoding the newly sampled points \mathbf{z} , along with their corresponding elastic surfaces computed by FEM homogenization. We observe that randomly sampled points in the neighborhood of the starting point exhibit similar topological features and mechanical properties. This similarity can be attributed to the property predictor that aims to learn the mapping from the latent space to properties. By incorporating physical knowledge into the VAE model, the predictor encourages the model to generate similar latent representations for trusses with similar properties for better prediction performance, as demonstrated in previous work with a VAE setup⁴ (albeit in a completely different context).

Abrupt changes in properties when interpolating between points in the latent space (as pointed out by the reviewer) can occur for several reasons. Figure 3b of the manuscript presented examples of interpolating between two structures

Figure 3. Representations of sampling results in the latent space. (a) Two-dimensional PCA analysis of latent space of the VAE model jointly trained with the property predictor. Colors indicate the beam radius of the truss structure. (b) Illustration of points sampled in the vicinity of the starting point ($\beta = 0.05$). (c) Representative examples of truss structures and their corresponding elastic surface generated by decoding from randomly-sampled points from the latent space. Highlighted in red is the starting point in part (a) and (b).

exhibiting *significantly different* mechanical properties in terms of the largest and the smallest Young’s modulus E_{11} and the universal anisotropy index A^U . Therefore, the start and end points in those examples are expected to reside in distinct regions in the latent space. In addition, we use 0 or 1 to denote the presence or absence of beam connections in truss structures, which poses a challenge for interpolation since intermediate values generated during interpolation lack physical interpretation. To avoid such artifacts, we threshold the outputs to binary values during post-processing. This, however, may lead to sudden changes in properties due to the inherent sensitivity of truss lattice properties to topological and geometrical features. Hence, abrupt changes may also be an artifact of how we interpret the predicted structures on a practical level. To mitigate this issue, instead of the binary representation of truss topology, one can adopt a continuous parameter $\lambda \in [0, 1]$ as a scaling factor of the beam radius, where intermediate values represent different strut diameters (i.e., each strut could have its own diameter) in analogy to the fading between pixel values in images (this was not considered in our study).

Figure 3a of the manuscript and Movies S1-S3 presented examples of truss structures generated by traversals along different latent axes. Specifically, we randomly pick a truss structure, map it to the latent space to obtain its latent representation, and then independently change each of the latent variables (which correspond to topology-specific, geometry-specific, and shared dimensions in the latent space, respectively) along their negative and positive directions with a fixed step size to showcase the transformations encoded in different latent dimensions. Our goal here is to demonstrate that our decomposition approach achieves the disentanglement of different structural components and enables controllable generation of truss structures without increasing model complexity. The transition between properties may not be smooth as the interpolation follows a straight line between points (which is different from the actual underlying manifold of the data) and may result in jumps or shifts in the latent space. To achieve smoother and more consistent transitions in properties of generated outputs, one can, e.g., sample points in the vicinity of those that exhibit desired properties to maintain similar behavior, or adopt a smaller interpolation step size, which would lead to more gradual and refined changes in properties.

Let us close by explaining the *“human interpretation”* of the latent space. One of the limitations of VAEs is that the latent variables often exhibit entangled behavior due to the complexity of the input distribution and the limited capacity of the model. In other words, different dimensions of the latent space are not independently controllable, which is undesirable in the generative process since variations in one latent axis may lead to unintended modifications of multiple features of generated outputs. To address this issue, we modified the structure of the VAE model and decomposed the latent space dimensions into topology-specific, geometry-specific, and shared dimensions. Therefore, different axes of the latent space encode unique feature transformations of trusses. The resulting structured latent space enables controllable generation of novel structures by manipulating various features of truss structures, as shown in Figure 3a of the manuscript. (In other words, we can independently choose to vary only the node positions or the connectivity or both, when moving in the latent space.) By decomposing different factors of variation while preserving the explicit interdependence between factors, the model captures the underlying relationships between these entities more effectively by exploiting the decomposed representations. This decomposition not only enhances the interpretability of the model¹² but also improves its ability to extrapolate beyond the training distribution¹³. Overall, the above characteristics of the latent space make it a meaningful and structured representation of the original discrete design space, which is advantageous for the efficient generation and optimization of novel truss structures with targeted properties.

Reviewer 1.4: *The definition of the “maximal” topology is rather arbitrary, some existing works consider less vertices, but maybe even more fine-grained topologies would enable even more extreme properties?*

Response 1.4: We agree that further exploration of other topologies could be valuable. However, our intention is to introduce a general and compact graph-based parameterization of truss lattices that enables efficient optimization and exploration of their properties, which can be extended to other topologies as well.

The reviewer makes a valid point that truss structures, including those with fewer (different) vertices or with more fine-grained structures, could potentially exhibit more extreme properties. After all, the maximum number of vertices and of connections in any such approach are ad-hoc choices influenced by manufacturability and computational costs. However, the graph parameterization we have proposed here offers significant advantages in this regard. For example, the simple cubic lattice, which is the stiffest along all three principal directions at fixed density, is a subset of the design space presented in this work (but not of other approaches with, e.g., fixed numbers of vertices). The graph parameterization provides great flexibility and can be well generalized to represent other truss topologies in an analogous manner, e.g., by adjusting the number of vertices, defining different structures in each octant, and introducing additional design parameters such as varying strut diameters and base material properties, etc.

Indeed, exploring different truss topologies could unveil new, previously unexplored properties or enable the discovery of even more extreme material behaviors. The generative modeling approach we presented can be readily extended by adapting the specific design parameters.

Reviewer 1.5: *Furthermore, here lattices with constant strut diameters are considered, but couldn't varying or graded struts also be considered, see e.g., <https://doi.org/10.1002/nme.6869>, which would directly lead to a continuous topology parameterization? Thus the claim of the present graph representation being "generalizable and unified" is over-emphasized in my opinion.*

Response 1.5: We kindly disagree with the reviewer and even argue that it is the state-of-the-art design spaces that have been extremely limited but over-emphasized in the literature. While the varying strut thickness in ref.¹⁴ can easily be included in our approach, it is not possible to extend the approach in ref.¹⁴ to the diverse topological design space of our framework. First, the design space of ref.¹⁴ contains only 7 unique topologies (created by superposition of simple cubic, face-centered cubic, and body-centered cubic lattices) with only three geometrical parameters (strut thickness of each of the three lattices in the superposition). In contrast, our graph representation greatly surpasses the representation of ref.¹⁴ in the design of truss lattices – our training dataset contains 10,211 different topologies and 965,736 unique geometries. The resulting design space (which is not restricted to those topologies but may add more vertices as well as add/remove connections when sampling from the latent space) is considerably larger. The 7 unique topologies of ref.¹⁴ are only a very small subset of our design space. The design spaces as in ref.¹⁴ are restricted to the chosen (extremely small) catalog of base unit cells, so that any generated design depends dramatically on the choice of the catalog without knowing the implications and restrictions. Second, we stress that their design space can indeed be expressed using the graph-based representation in our work, simply by including the varying strut diameters and material properties as edge features.

Minor comments:

Reviewer 1.6: *In Fig. 1a+b, it would be helpful to consistently color the nodes as vertex, edge, face & body nodes.*

Response 1.6: We thank the reviewer for the helpful feedback and have updated the figure in the revised manuscript accordingly to enhance the clarity and readability of the figure.

Reviewer 1.7: *On page 3, the lattices and elastic tensors are referred to as generally "anisotropic", but I believe it would be better to directly mention the symmetry, which should be rhombic in my opinion.*

Response 1.7: We have revised the manuscript to specify that the lattices and elastic tensors exhibit *orthotropic* symmetry.

Reviewer 1.8: *The correlation plots for assessing the performance of the VAE look generally good, but Fig. S4a-y shows a few extreme outliers. What might be the reason and how could this be improved? What are the consequences of such outliers?*

Response 1.8: We thank the reviewer for raising this point. In response, we have added a brief note discussing the extreme outliers in Section 3.3 of the revised Supplementary Information.

Let us examine here the two extreme outliers in Figure S4a-y (correlation plot of true vs. VAE-reconstructed y -coordinate nodal positions). In Figure 4 below (which has been added to the Supplementary Information), we show the comparison of the reconstructed vs. true structure, and their corresponding elastic surfaces of the two outliers of Figure S4a-y. We observe that, although there is a significant difference in the reconstructed vs. true node position of structure (b) in Figure 4a, the difference in their effective structure and properties is minor. In addition, Figure S4 in the supplementary information shows that the reconstructed node positions correlate well with the ground truth with R^2 -values (≈ 0.999) close to unity for each component. Over 99.9% of all instances in the test data had errors within the interval $[-0.048, 0.048]$ for component $y \in [0, 1]$, indicating that the overall reconstruction performance is satisfactory. Therefore, we are confident that the VAE obtains a reliable estimation of the intrinsic underlying distribution and provides a sufficiently good reconstruction of various truss structures.

Despite near-perfect overall accuracy, there are several potential reasons for the presence of extreme outliers in the correlation plots. One possible explanation is the trade-off between loss terms in this multi-task learning problem.

Specifically, the VAE model is jointly trained with the property predictor as

$$\theta, \phi, \omega \leftarrow \arg \min_{\theta, \phi, \omega} \underbrace{\frac{1}{N} \sum_{n=1}^N \left(\left\| \mathbf{A}^{(n)} - \mathbf{A}^{(n)'} \right\|^2 + \left\| \mathbf{x}^{(n)} - \mathbf{x}^{(n)'} \right\|^2 \right)}_{\text{reconstruction loss}} + \underbrace{\frac{1}{N} \sum_{n=1}^N \left\| \mathbf{S}^{(n)} - \mathcal{F}_\omega[\boldsymbol{\mu}^{(n)}] \right\|^2}_{\text{property prediction loss}} + \underbrace{\sum_{n=1}^N D_{\text{KL}} \left(\mathcal{N} \left(\left[\mu_1^{(n)}, \dots, \mu_d^{(n)} \right]^\top, \text{diag} \left(\left[\sigma_1^{(n)2}, \dots, \sigma_d^{(n)2} \right]^\top \right) \right) \parallel \mathcal{N}(\mathbf{0}, \mathbf{I}) \right)}_{\text{Kullback-Leibler divergence}}, \quad (2)$$

where θ, ϕ, ω are model parameters of the VAE. Minimizing multiple loss functions simultaneously can be challenging because, e.g., the model capacity is limited, or some loss terms are in conflict such as the inherent trade-off between the reconstruction robustness and generalization (given by the Kullback-Leibler divergence) in VAEs. A common strategy is to balance the losses as weighted sums of related terms during the multi-objective optimization, whereas the choice of weights will have a strong impact on the performance of the model on different tasks. In this work, we have compared the model performance with several selections of weights and selected the best model based on the property prediction accuracy, because our focus is the downstream task of optimizing truss lattices for target properties.

Alternatively, one can choose the best model according to other criteria depending on actual goals. For example, we modify Equation 2 to include a weight $\lambda_{\text{recon}} > 0$ for the reconstruction loss as follows:

$$\theta, \phi, \omega \leftarrow \arg \min_{\theta, \phi, \omega} \lambda_{\text{recon}} \underbrace{\frac{1}{N} \sum_{n=1}^N \left(\left\| \mathbf{A}^{(n)} - \mathbf{A}^{(n)'} \right\|^2 + \left\| \mathbf{x}^{(n)} - \mathbf{x}^{(n)'} \right\|^2 \right)}_{\text{reconstruction loss}} + \underbrace{\frac{1}{N} \sum_{n=1}^N \left\| \mathbf{S}^{(n)} - \mathcal{F}_\omega[\boldsymbol{\mu}^{(n)}] \right\|^2}_{\text{property prediction loss}} + \underbrace{\sum_{n=1}^N D_{\text{KL}} \left(\mathcal{N} \left(\left[\mu_1^{(n)}, \dots, \mu_d^{(n)} \right]^\top, \text{diag} \left(\left[\sigma_1^{(n)2}, \dots, \sigma_d^{(n)2} \right]^\top \right) \right) \parallel \mathcal{N}(\mathbf{0}, \mathbf{I}) \right)}_{\text{Kullback-Leibler divergence}}. \quad (3)$$

After training the model with a higher weight of the reconstruction loss term ($\lambda_{\text{recon}} = 5$, as opposed to $\lambda_{\text{recon}} = 1$ originally), Figure 4b shows that the points are more centralized around the line with unit-slope and zero-intercept with fewer outliers in the correlation plots of reconstructed vs. true node positions (evaluated on the same test dataset used in the manuscript), which, however, could lead to compromised or degraded property predictor accuracy due to the finite total capacity of the model, as shown in Table 2 below.

R^2 -scores	x	y	z	\mathbb{C}_{1111}	\mathbb{C}_{1122}	\mathbb{C}_{1133}	\mathbb{C}_{2222}	\mathbb{C}_{2233}	\mathbb{C}_{3333}	\mathbb{C}_{4444}	\mathbb{C}_{5555}	\mathbb{C}_{6666}
$\lambda_{\text{recon}} = 1$	0.999	0.999	0.999	0.995	0.987	0.989	0.995	0.990	0.995	0.983	0.983	0.982
$\lambda_{\text{recon}} = 5$	0.999	0.999	0.999	0.982	0.961	0.966	0.981	0.965	0.979	0.980	0.981	0.976

Table 2. Comparison of the node positions reconstruction accuracy of the VAE model and the prediction accuracy of the property predictor using $\lambda_{\text{recon}} = 1$ and $\lambda_{\text{recon}} = 5$.

Reviewer 1.9: *Furthermore, in Figure S5 especially the shear components are not approximated so well – what could be the reason? Maybe it would help to predict the Cholesky decomposition instead of the components of the elasticity tensor to preserve positive definiteness? See <https://doi.org/10.1016/j.jcp.2020.110072>, <http://arxiv.org/abs/2203.13938>*

Response 1.9: Figure S5 of our manuscript demonstrates that the predicted values from the property predictor are accurate with respect to the ground truth for each component, as indicated by the high R^2 -values (≥ 0.982). To elucidate the occurrence of the relatively lower accuracy of shear components, we analyzed the error distribution of the predictions for each component on the test set, as depicted in Figure 5. The error distributions for all components follow a zero-centered normal-type distribution with small standard deviations, indicating the absence of systematic sources of errors. It can be observed from Figure 6 that the shear components have smaller value ranges compared with other components, which may lead to a bias in the learning process. During training, the model might prioritize minimizing errors in dimensions with larger values, while neglecting dimensions with smaller values, thus leading

Figure 4. Evaluation of the reconstruction accuracy of the VAE model. (a) Illustration of selected extreme outliers from Figure S4a of truss structures and their corresponding elastic surfaces. (b) Reconstructed vs. true 3D components (x, y, z) of the node positions in the test dataset using $\lambda_{\text{recon}} = 5$. Dashed lines represent the ideal lines with zero-intercept and unit-slope; the corresponding coefficient of determination R^2 -scores are indicated.

to lower accuracy in shear components. To mitigate this issue, one can increase the weight of the loss for shear components, or normalize labels across all dimensions to a similar range. Additionally, the choice of the loss function can also play a role in handling differences in the range of values, for example, mean squared error (MSE) can be sensitive to outliers or large errors in dimensions with larger value ranges. In such cases, adding regularization terms or imposing weights for different dimensions in the loss function can be beneficial for the model's overall accuracy.

We appreciate the reviewer's suggestion to predict the Cholesky decomposition instead of the components of the elasticity tensor and have explored it with our framework. Specifically, the orthotropic stiffness matrix \mathbb{C} , being symmetric positive-definite, can be decomposed as $\mathbb{C} = \mathbf{L}\mathbf{L}^T$, where the Cholesky factor \mathbf{L} is a lower-triangular matrix with positive diagonal entries, i.e.,

$$\mathbf{L} = \begin{bmatrix} L_{1111} & & & & & \\ L_{2211} & L_{2222} & & & & \\ L_{3311} & L_{3322} & L_{3333} & & & \\ & & & L_{2323} & & \\ & & & & L_{1313} & \\ & & & & & L_{1212} \end{bmatrix} \quad (4)$$

with $L_{1111}, L_{2222}, L_{3333}, L_{2323}, L_{1313}, L_{1212} > 0$. In response to the reviewer's comment, we used Cholesky factors ($L_{1111}, L_{2211}, L_{2222}, L_{3311}, L_{3322}, L_{3333}, L_{2323}, L_{1313}, L_{1212}$) as the output of the property predictor (instead of the stiffness components directly) and trained the model jointly with the VAE. The prediction performance (evaluated on the reconstructed stiffness components) is shown in Figure 7 below. While the model shows superior accuracy of

the shear components ($\mathbb{C}_{2323}, \mathbb{C}_{3131}, \mathbb{C}_{1212}$) relative to the other components, the overall accuracy of all components is degraded compared to the result presented in our manuscript. Further adjustments to the model structure and tuning of hyperparameters may be necessary to enhance the overall performance.

In summary, significant differences in the range of values across different dimensions of the labels can potentially affect the accuracy of the model. Normalization techniques and appropriate choice of loss functions can help address this issue and improve the model’s performance. We agree that predicting the Cholesky decomposition can be beneficial when positive definiteness is a critical consideration, as demonstrated in ref.^{15,16} mentioned by the reviewer, which is indeed an area of interest and potential investigation. Yet, we demonstrated that it does not have a significant impact in our model.

Figure 5. Histograms of test errors observed in the prediction of the effective stiffness tensor components.

Reviewer 1.10: *Since the decoding accuracy is only 82%, what should one do when an optimization result cannot be decoded successfully?*

Response 1.10: The reviewer raises an interesting point regarding cases where an optimization result cannot be successfully decoded into a physically valid truss structure. Let us first explain the source of not having 100% validity score (which is distinct from the decoding/reconstruction accuracy) before we present our remedy.

While the careful construction of our VAE architecture ensures that the decoded graphs (i.e., node positions and connectivity) are identically valid, physical constraints – such as the generated structure must be singly connected or every node must have at least two connected beams – are not enforced by construction and can still lead to

Figure 6. Histograms of the effective stiffness tensor components.

Figure 7. Evaluation of the prediction accuracy of the property predictor model using Cholesky decomposition. Predicted vs. true components of the stiffness tensor \mathbb{C} in the test dataset (same as in our manuscript). The model property predictor predicts components of the Cholesky factor \mathbf{L} , and the predicted stiffness is computed by $\mathbb{C} = \mathbf{L}\mathbf{L}^T$. All dashed lines represent the ideal line with zero-intercept and unit-slope; the corresponding R^2 -deviations are indicated.

mechanically invalid truss structures when decoding randomly sampled points from the latent space.

To mitigate this issue during design optimization in the latent space, we selected the 100 different initial guesses and ran the optimization in parallel. The high validity score of 82% means that on average 82 out of 100 of those optimization runs result in a physically valid truss lattice.

While it is possible to impose additional constraints on the VAE model to restrict it to generating only valid structures, we chose to employ this post-processing step as it keeps the VAE structure simple without adding significant additional computational expense, while yielding valid structures 82% of the times which is more than sufficient for our purposes.

Reviewer 1.11: *Page 9, line 2: A number is missing after “Section”*

Response 1.11: We thank the reviewer for pointing this out. We have corrected the missing number in the sentence.

Reviewer 2:

The manuscript ‘Unifying the design space of truss metamaterials by generative modeling’ is well written and novel piece of work. This paper provides a graph-based deep generative network that combines a variational autoencoder and a structure-property predictor. The discrete design space for trusses is reduced to continuous latent space, thus enabling optimization in a continuous space to obtain truss structures with desired properties.

The reviewer recommends publication of the article after the following queries are addressed satisfactorily:

Response 2.0: We are grateful to the reviewer for their positive evaluation and gladly respond to the reviewer’s queries in the following.

Reviewer 2.1: *It is quite innovative to decompose the latent space dimensions into topology, geometry, and their shared dimensions. However, it would be nice to spell out the procedure of the decomposition in relatively more details. Also, it is not entirely clear how the dimensions of different subspaces, such as topology-, geometry-specific spaces, are determined.*

Response 2.1: We thank the reviewer for raising this point. In response, we have added more comprehensive descriptions of the decomposition process and its implementation in Section 3.2 of the revised Supplementary Information.

In order to accommodate the interdependencies between the structural topology (represented by the adjacency matrix) and node placements (represented by node features), we modify the VAE structure by restricting which part of a latent embedding is used for each task. Let $d_A - d_{Ax}$ be the number of latent dimensions allocated exclusively to topology, $d_x - d_{Ax}$ the number of dimensions exclusively to geometry, and d_{Ax} the number of dimensions allocated to both, as illustrated in Figure 2 above. To keep the overall number of trainable parameters fixed, we reduce the outputs of the encoder ($\mu^A, \log \sigma^A \in \mathbb{R}^{d_A}$, and $\mu^x, \log \sigma^x \in \mathbb{R}^{d_x}$) to obtain the final latent representations $\mu, \log \sigma \in \mathbb{R}^d$ with $d = d_A + d_x - d_{Ax}$ by as

$$\mu = \mu_{1:d_A-d_{Ax}}^A \oplus \frac{1}{2}(\mu_{d_A-d_{Ax}+1:d_A}^A + \mu_{1:d_{Ax}}^x) \oplus \mu_{d_{Ax}+1:d_x}^x, \quad (5)$$

$$\log \sigma = \log \sigma_{1:d_A-d_{Ax}}^A \oplus \frac{1}{2}(\log \sigma_{d_A-d_{Ax}+1:d_A}^A + \log \sigma_{1:d_{Ax}}^x) \oplus \log \sigma_{d_{Ax}+1:d_x}^x, \quad (6)$$

where \oplus denotes vector concatenation. Our rationale is that the topological and geometrical features of truss lattices are strongly correlated and, therefore, the overlapping dimensions should learn to extract the shared information between different entities, while preserving the topology-specific and geometry-specific information within their respective dimensions of latent representations. Compared to the non-overlapping model, where shared information is stored redundantly, the overlapping model, with its ability to capture joint information of different features allows us to adjust the importance given to their interdependencies, while maintaining a constant number of training parameters.

The process of choosing the dimensionality of the latent space is discussed at length in Response 1.2 above and in Section 3.2 of the revised Supplementary Information (in a nutshell, it is determined empirically).

Reviewer 2.2: *The framework seems to be general enough to be applicable to different kinds of metamaterials, such as plate- and shell-lattices. However, the method demonstrated here for truss-lattices. Can the framework be extended to other kind of lattices, for example shell-lattices? If not, could the author suggest the required modifications that are necessary to the framework?*

Response 2.2: We thank the reviewer for this insightful comment. The reviewer is correct that this framework has the potential to be applicable to other types of metamaterials. In this work, we developed a graph-based deep learning approach exploiting the resemblance between truss lattices and computational graphs, where the struts and their intersections can be naturally translated into edges and nodes in graphs. In fact, the fundamental theory and methods outlined in this paper can easily be adapted to other structures by modifying the specific parameterization and representation of (meta-)materials of interest. For example, ref.² employed graph representations for shell lattices and developed surrogate models for predicting their effective properties based on graph neural networks. The utilization of graph-based methods has also been demonstrated in the study of polycrystalline materials¹⁷ and origami-inspired metamaterials¹⁸, where our framework can still apply by adapting the corresponding graph parameterization. Besides graph representations, the generative modeling framework can also be extended to other families of metamaterials, including 2D pixelated metamaterials^{19,20}, 3D spinodal metamaterial²¹, and 3D kirigami metamaterials²² by utilizing appropriate neural network architectures as the encoder and decoder, e.g., convolutional neural networks for pixelated/voxelized inputs or fully connected neural networks for inputs represented as vectors. This highlights the potential of our framework to analyze and optimize for a broader range of metamaterials. We have added a brief note on this in the *Discussion* section of the revised article.

Reviewer 2.3: *The author focused only the elastic behavior of the truss metamaterials. However, it appears that just by modifying the structure-property predictor, it might be possible to optimize for the other properties, such as toughness, dispersion relations, as well. If this is correct, could the authors provide a brief discussion about the plausibility of extending the work for simultaneously optimizing two or more properties?*

Response 2.3: The reviewer makes a good point that modifying the structure-property predictor may allow for the optimization of additional properties beyond the elastic behavior with proper parameterization approaches. Recent works^{23,24} have investigated the structure-to-property mapping of the dispersion relations of metamaterials using deep learning approaches. Our framework can be adapted to investigate other properties by modifying the target of the structure-property predictor with a similar setup as in the above works. In response to the reviewer’s comment, we have extended our approach beyond the linear regime to the inverse design of periodic trusses with target nonlinear responses to showcase the generalization ability of the proposed generative modeling framework, as demonstrated in Figure 5 of the revised article (reproduced here in Figure 1).

To extend our framework to the simultaneous design of multiple properties, one possible approach is to develop a multitask property predictor. This can be achieved by feeding the extracted features from intermediate layers to the separate heads of the final output layer to enable simultaneous prediction for various material properties, as demonstrated in Figure 8 below. Multi-task learning, combined with graph neural networks, has shown promising results in jointly predicting multiple properties of molecules²⁵, ferromagnetic materials²⁶, and crystalline materials²⁷. By extending our model to include separate output heads for each property of interest and training the model jointly on a dataset encompassing various desired properties, we can effectively leverage the correlations and shared information among different targets, which enables the simultaneous optimization of various properties by integrating the multi-task property predictor into a multi-objective optimization framework. We have added a brief note on this in Section *Generative modeling framework* of the main article.

Reviewer 2.4: *As have been mentioned by the authors, there are multiple truss structures that have similar properties. However, in Figure 4, which demonstrates the optimal truss structures for desired properties, shows only one such optimal lattice structure. It would be nice to provide the details, possibly in the supplementary, as to how the initial structures are chosen, their properties at different optimization iterations and some of the optimal lattice structures that have similar properties.*

Response 2.4: We thank the reviewer for their valuable comment and have added more details of the optimization procedure in Section 3.5 of the revised Supplementary Information. Let us provide the main points here.

For each optimization task, we first evaluate structures in the training dataset and select the 100 closest matches in terms of the target property as the initial guesses. Then, we perform gradient-based optimization for each initial guess in parallel and identify the optimal solution followed by FE homogenization-based validation of their properties. As the reviewer suggested, we have added visualizations of truss structures obtained along the optimization path of

Figure 8. Schematic illustration of the multi-task learning framework. The feature extraction layer extracts task-specific representations from the latent space, which are then fed into separate branches corresponding to different prediction tasks.

minimizing Poisson’s ratio ν_{21} as a representative example in Figure 9a below, which showcases the evolution of the geometries and properties.

To further emphasize that multiple truss structures exhibit similar properties, we have sampled in the neighborhood of the optimal solution based on Equation 1 (see also Response 1.3). Figure 9b shows that novel structures generated by sampling in the neighborhood of a point in the latent space exhibit similar topological features and mechanical properties (see also Figure 3 above). Notably, some structures generated through sampling in the latent space even display superior target properties compared with the solution we obtained through optimization, which highlights the potential of our generative framework in not only memorizing the training data but generalizing from a comprehensive truss dataset, effectively capturing the underlying mechanical relations. This finding further reinforces the notion that our generative framework is highly structured and is capable of leveraging the encoded mechanical property information within the latent space.

We thank the reviewer for raising this point. Those newly added details and visualizations indeed offer a more comprehensive understanding of the inverse design framework and showcase the capabilities of our approach in exploring and discovering diverse truss structures with unprecedented properties.

Reviewer 2.5: *The authors leverage the graph representation of the truss based lattices. Have the authors considered permutation and equivariance for these graph representations i.e. if a lattice is rotated in the Euclidean space, will the graph interpret the two lattices as identical or different?*

Response 2.5: This is a good point! We appreciate the opportunity to further clarify these aspects.

Permutation invariance is indeed a desirable property in graph representations as it ensures that the graph neural network recognizes the inherent symmetry in the graph, which has proven to be efficient in the supervised learning of permutation-invariant graph-level properties, e.g., the structure-to-property mapping¹⁷. In our work, however, the graph representations do not inherently possess permutation invariance due to the complexity arising in unsupervised learning tasks. In general, a graph with n nodes can be equivalently represented by up to $n!$ adjacency matrices, each corresponding to a different order of nodes. Such high representation complexity poses a challenge in learning for generating permutation-invariant graphs, and as a result, most of the existing generative models for graphs are not invariant to the chosen ordering²⁸ due to the ambiguous reconstruction objective; i.e., the model needs to know which input node corresponds to which output node in order to compute the reconstruction loss, which leads to an expensive graph matching problem to solve^{29,30}. To simplify the process, we chose one permutation of nodes and edges, and the decoder generates the graph structure in a fixed order.

Regarding equivariance to rotation and translation, we here focus on the generative modeling of truss structures without assuming specific spatial orientations and do not explicitly consider these properties in our approach. In the

Figure 9. Visualizations of truss structures based on gradient optimization to minimize Poisson’s ratio ν_{21} . (a) Evolution of truss geometries and their elastic stiffness properties along the optimization path. (b) Representative examples of truss structures obtained by sampling in the vicinity of the optimal solution in (a) with the sampling factor $\beta = 0.05$ using Equation 1. The shown elastic stiffness surfaces were obtained by FE homogenization.

context of our study, we aim to predict the homogenized stiffness tensor components from the latent representation of truss lattices, which do not exhibit 3D translation and rotation symmetries, and therefore, the resulting graph representation will be different from the original one if a lattice is rotated in the Euclidean space. In some other domains, however, it is important to enforce equivariance/invariance to 3D transformations in graph models, such as molecular property prediction^{31,32} and complex physical systems modeling³³.

In summary, the graph representations used in our work do not possess permutation invariance or equivariance to rotation and translation, given the complexity arising in the generative modeling of graphs. Exploring permutation invariance and equivariance in graph generation^{34–36} presents an interesting avenue for future research, particularly in scenarios where the same output is expected from the network regardless of permutation or spatial transformations of the truss lattice, such as predicting scalar labels like the energy^{37,38}.

Minor comments:

Reviewer 2.6: *The authors have mentioned that spherical interpolations in Eq. (6). However, it is not clear whether linear or spherical interpolations are used in interpolations in the latent space, for example in Fig. 3. If spherical interpolations are used, it is not entirely clear why such interpolations, instead of linear interpolations, are necessary.*

Response 2.6: We appreciate the reviewer’s valuable feedback. To clarify, the truss structures presented in Figure 10a (reproduced from Figure 3 of the manuscript) were generated by traversals along different latent axes, which involves randomly selecting a truss structure, mapping it to the latent space to obtain its latent representation, and independently modifying each of the latent variables (corresponding to topology-specific, geometry-specific, and shared dimensions in the latent space, respectively) along their negative and positive directions with a fixed step

size. When interpolating between two points whose corresponding trusses exhibit extreme mechanical properties (as illustrated in Figure 10b), we employ *spherical linear* interpolation (slerp) rather than linear interpolation (lerp) in the latent space.

The choice of slerp over lerp is motivated by several factors. Constructing an approximately continuous latent space of truss structures provides a significant advantage, as the vector representation of truss lattices allows for the generation of novel structures by arithmetic operations. However, performing interpolation in a high-dimensional latent space with a Gaussian prior presents challenges. First, lerp assumes a straight line between points, which ignores the underlying structure of the data distribution, as shown in Figure 11a below. Additionally, lerp measures the Euclidean distance between two points, which, however, does not necessarily indicate the similarity between truss structures in high-dimensional spaces. Consequently, the linearly interpolated points may jump or traverse regions of the latent space with similar representations, leading to inconsistent or unrealistic interpolations. In contrast, slerp takes into account the spherical structure of the latent space and follows the shortest arc on a hypersphere, which reduces artifacts or unnatural transitions between points. Figure 12 shows that linear interpolation produces less coherent geometries with abrupt changes in properties, whereas slerp interpolation (as shown in Figure 3b in the manuscript) exhibits smoother transitions in both geometries and properties along the interpolation path. The effectiveness of slerp has been demonstrated in the context of various generative models^{4,39} with uniform and Gaussian priors.

Overall, slerp is generally more suitable for generating smooth and meaningful interpolations in the latent space of generative models. We thank the reviewer for the comment and have added a discussion in Section 3.4.2 of the revised Supplementary Information to provide a clearer explanation of the interpolation process and the rationale behind the selection of slerp.

Reviewer 2.7: *On Page 9, the sentence “To this end, we adopt the joint embedding model presented in Section to encode the topological and geometrical features in different dimensions of z , while maintaining the total number of latent dimensions constant.” seems to have a missing section reference.*

Response 2.7: We thank the reviewer for pointing this out and have corrected the sentence in the revised manuscript.

Reviewer 2.8: *Are the elastic surfaces in Fig. 3 predictions or ground truth from numerical calculations?*

Response 2.8: The elastic surfaces of generated samples were obtained by FE homogenization simulations to demonstrate the changes in the mechanical properties along the interpolation path in the latent space. The property predictor generalizes well on these interpolated structures (averaged root mean square error of 0.001). The prediction performance of the model on unseen structures will be further discussed in Response 2.10. We thank the reviewer for raising this point and have revised the manuscript to provide further clarification.

Reviewer 2.9: *On Page 4, the authors mention “(e.g., the representative examples shown in Figure 1d reach effective Young’s moduli of ca. 36% higher than that of a simple cubic unit cell in the principle direction at the same density)”. Can the authors clarify that this is possibly true for just one direction.*

Response 2.9: The reviewer is right: the stiffening in one direction comes with softening in the other two. This is also visible in Figure 1d (left figure), which includes the projections of the dataset, including the simple cubic unit cell, onto the Young’s moduli E_{11} -, E_{22} -, and E_{33} -planes. We have added a comment to the revised manuscript to clarify this point.

Reviewer 2.10: *On Page 7, the authors mention “Results show that on average 82.3% (evaluated on 1,000 attempts of random sampling) of randomly-selected samples can be successfully decoded into valid (i.e., physically meaningful) truss topologies – we refer to this fraction as the validity score”. What about the accuracy of stiffness prediction on those?*

Response 2.10: We thank the reviewer for raising this point. To provide further clarification, we present the root mean squared error (RMSE) of the predicted stiffness values with the ground truth (obtained by FE homogenization) for the randomly sampled structures (of which on average 82.3% structures will be valid) in Table 3 below. We note that the error of random samples is generally higher than that of the test dataset, but still within the acceptable regime. One possible reason for the reduction in accuracy is that these structures are generated by random sampling from the prior, i.e., from a multivariate Gaussian distribution, which may not perfectly align with the constructed latent space. Consequently, randomly generated samples may fall into data-sparse regions in the latent space. To enhance the model’s performance on random samples, one can augment the training dataset with novel structures

Figure 10. Representative examples of interpolation in the latent space (reproduced from Figure 3 of the manuscript). Samples are generated by (a) traversals along three different latent axes: (1) taken from the topology-specific, (2) shared topology and geometry, and (3) geometry-specific dimensions of the latent space; (b) interpolation between two points in latent space, whose corresponding trusses exhibit extreme mechanical properties (in terms of directional Young's modulus E_{11} and the universal anisotropy index A^U). Their corresponding 3D elastic surface evolution (obtained by FE homogenization) is shown along the interpolation path.

Figure 11. Illustration of the lerp and slerp interpolations with interpolation parameter $\alpha \in \{0.25, 0.5, 0.75\}$. (a) *Linear* interpolation: interpolated vectors are obtained by $\text{LERP}(z_1, z_2; \alpha) = \alpha \cdot z_1 + (1 - \alpha) \cdot z_2$. (b) *Spherical* interpolation: the interpolation is based on the angle between two points, and the resulting interpolation path follows a great circle on the hypersphere.

Figure 12. Representative examples of linear interpolation in the latent space. Samples are obtained by linear interpolation between two points in the latent space, whose corresponding trusses exhibit extreme directional Young’s modulus E_{11} . Their corresponding 3D elastic surface evolution is shown along the interpolation path.

decoded from randomly sampled points and then fine-tune the property-predictor while keeping the encoder and decoder fixed. This is an interesting avenue for future exploration to further improve the model’s robustness and generalization capabilities.

RMSE	C_{1111}	C_{1122}	C_{1133}	C_{2222}	C_{2233}	C_{3333}	C_{2323}	C_{3131}	C_{1212}
Random samples	1×10^{-3}	1×10^{-3}	9×10^{-4}	1×10^{-3}	1×10^{-3}	9×10^{-4}	9×10^{-4}	9×10^{-4}	9×10^{-4}
Test dataset	5×10^{-4}	3×10^{-4}	3×10^{-4}	5×10^{-4}	3×10^{-4}	5×10^{-4}	3×10^{-4}	3×10^{-4}	3×10^{-4}

Table 3. Comparison of the property prediction performance of the property predictor on randomly sampled structures and the test dataset.

Reviewer 3:

In summary, this manuscript is well-written and could be accepted after several minor revisions. Here are the comments:

Reviewer 3.1: *In the second part of Results, it is mentioned that the proposed deep learning method can avoid the expensive computation cost of the FE homogenization computation. However, neural network surrogate model is still based on the finite element simulation. How to define this cost for the proposed method?*

Response 3.1: In Table S3, we present the computational cost and run times for the dataset generation, training, and inference/prediction stage. We agree that the neural network surrogate model is indeed based on finite element simulations during its training phase. However, we would like to point out two key advantages.

(i) The training time cost is only incurred once (not to mention that such simulations for data generation are ideally run in parallel). After offline training, the property-predictor neural network can be used to instantly obtain the property without the need for further expensive finite element computations to evaluate the effective material response of truss lattices. This means that the computational cost associated with our method is reduced by orders of magnitude during the property evaluation phase in downstream tasks such as searching for structures beyond the training domain, with a possible extension to multiscale optimizations, e.g., towards designing spatially-variant microstructures, where nested FE evaluation could lead to intractable computational costs.

(ii) The design space is extremely high-dimensional. Even a million datapoints cover only sparsely this space due to the *curse-of-dimensionality*. E.g., a 6-dimensional space sampled sparsely with 10 points along each dimension will result in a dataset of size 10^6 . In our context, we have 278 binary and 27 continuous dimensions and a dataset size of only 965,736. Therefore, the ability to accurately predict properties and perform generative design in such a large design space significantly outweighs the cost of generating the dataset.

Reviewer 3.2: *The effective stiffness-related 3D coordinate labels in Figures 2 and 3 are not clear. The involved figures should be further improved.*

Response 3.2: We thank the reviewer for raising this point. We have modified the figures to improve their clarity.

Reviewer 3.3: *What are the criteria for classifying the five elementary truss lattices in Section 1 of Supplementary Information? Some necessary explanation should be given.*

Response 3.3: It is important to highlight that the chosen five elementary truss lattices are only the starting point for dataset generation and hence play only a minor role in the overall exploration of the design space (which allows for the insertion of new vertices and the removal of vertices, the addition/removal of strut connections, etc.). This is a key advantage of our approach over prior works that used a truss catalog without generalization capability^{9,14}. For the starting point, we selected five well-studied topologies (including the octet, simple cubic, and body-centered cubic) as a $1 \times 1 \times 1$ or $2 \times 2 \times 2$ tessellations, which span a wide range of elastic properties (from isotropic to strongly orthotropic). We have added the node positions and connectivities of the five considered elementary truss types in Figure 13 below and in Section 1 of the revised Supplementary Information. Starting from the elementary truss set, we randomly perturbed both the connectivities and the node positions, which greatly expanded the design space of achievable material properties. To further enhance the diversity of the dataset, it is possible to enrich the elementary truss set with more topology types such as diamond or kagome. Besides, considering the tessellation as an additional design parameter could lead to even more variations in the dataset. By perturbing the connectivity of these elementary trusses, we have effectively explored a wide range of available truss topologies and constructed a much more diverse truss set, as shown in Step 2 in Figure S1. Hence, we chose to restrict ourselves to the above five elementary trusses to strike a balance between diversity and computational efficiency.

Reviewer 3.4: *The meaning of “ r_{\max} ” in the conditions for adding new nodes mentioned in the Section 1 of Supplementary Information is also need to be explained.*

Response 3.4: We are grateful to the reviewer for pointing out this inconsistency in our description. r_{\max} refers to the maximum permissible length of a beam connection in the truss structure. When perturbing the truss connectivities, we enforce the constraint that added beams must not exceed r_{\max} to prevent excessively long connections and coinciding edges. For example, if V_0 is the midpoint node of V_1 and V_2 , graph edges $(V_0, V_1), (V_0, V_2)$ can be introduced, while the direct connection (V_1, V_2) is forbidden since it might overlap with the first. We chose the value of r_{\max} according to the longest connection in the initial five elementary trusses, which is $\sqrt{3}/2$ (times the unit side length of the unit cell). The value of r_{\max} given in Section 1 of the Supplementary Information was unfortunately incorrect. We apologize for the mistake and any confusion it may have caused. We thank the reviewer for bringing this error to our attention and have corrected it in the revised manuscript.

Reviewer 4:

The authors have described a very good and interesting topic. They have proposed a graph-based deep learning generative framework, and present a VAE model containing an encoder and a decoder to optimize by neural network surrogate model without costly FE homogenization computation. Thereafter, the authors have assessed the accuracy and flexibility of the VAE model. Finally, they study the possibility of using VAE model to design new configurations based on the latent space and the gradient-based optimization in the latent space. Overall, this work is of significance

Figure 13. Node positions \mathbf{x} and connectivity \mathbf{A} of the five considered elementary truss unit cells.

to the related fields. However, to allow the submitted manuscript to be accepted by such a well-renowned journal, some comments should be further concerned by the authors:

Response 4.0: We thank the reviewer for their helpful suggestions, which we have addressed as follows.

Reviewer 4.1: *To allow the work to be reproduced, some important details (e.g., axial stiffness and length of the members) should be provided.*

Response 4.1: We appreciate the reviewer’s suggestion regarding the provision of additional details to facilitate reproducibility. We have added the node positions and connectivities of the five considered elementary truss types in Figure 13 and in Section 1 of the revised Supplementary Information. The full details of all structures, including the truss structure dataset and the corresponding effective stiffness values, are made freely available with the published manuscript to enhance the transparency and reproducibility of our research. As the elastic stiffness is scale-invariant (in the absence of material size effects), the unit cell’s length and axial strut stiffness were taken as unity (all other quantities scale relative to those).

Reviewer 4.2: *Both graph theory (with nodal positions considered, i.e., gemetric-graph-theoretic) and ML are adopt in this study. In fact, it would be much better if the graph thoery is combind with symmetry (or group theory).*

Response 4.2:

The reviewer raises a good point. We have added a comment in the revised manuscript to provide further clarification. We agree that combining graph theory with symmetry or group theory has the potential to yield even more comprehensive and powerful analyses by identifying equivalent structural configurations, e.g., isomorphic (i.e., topologically identical) graphs should lead to the same predicted material properties, which in turn are frame-indifferent^{17,40}. In this paper, we did not leverage the permutation invariance or equivariance of the graph representation due to the inherent representation complexity in graph generative models, as discussed in Response 2.5. However, recent permutation-invariant and 3D rotation/translation equivariant graph neural networks have demonstrated their efficiency in a variety of applications^{37,41–43}, and we believe that incorporating symmetry groups such as SE(3) in our graph generative modeling framework could indeed enhance the learning of the underlying mechanics of truss structures. This is an interesting potential avenue for future research – we hope the reviewer

understands that it goes beyond the scope of this study.

Reviewer 4.3: *This reviewer is very interested in whether the RVW has been selected under relatively fair conditions (such as the same density, same volume, same mass) during the ML process.*

Response 4.3: In our study, we did aim to select truss structures under fair conditions to ensure a consistent and unbiased comparison of the effective material properties. When constructing the training dataset, we used the same dimensions of all unit cells, while the strut diameters were uniformly scaled such that a constant relative density of $\rho = 0.15$ was obtained across all structures. All reported effective stiffness values are obtained from FE homogenization, assuming a solid base material with Poisson's ratio $\nu_s = 0.3$ and unit Young's modulus $E_s = 1$ (as the response is scale-invariant and scales linearly with E_s). These details are provided in the Section *Creating the design space* of the manuscript.

Reviewer 4.4: *As far as engineering applications are concerned, the authors should consider potential difficulty of the connectivity of the joints, and potential local buckling of the slender members under compression.*

Response 4.4: The reviewer raises several interesting points regarding the applicability of our study to engineering applications. In our study, we aimed to construct a diverse set of truss lattices that encompasses a wide range of material configurations and property variations (in fact, wider than any other study before). Therefore, we did not consider specific manufacturability constraints (even though we do not foresee any significant challenges in additively manufacturing the presented truss lattices with struts of finite thickness and length by common techniques such as SLA or TPL). It is possible to enforce specific constraints to narrow down the design space depending on the goals of specific applications. For instance, geometric constraints can be imposed to limit the maximum size or number of edges of the truss graph; printability constraints⁶ can be considered by ensuring that every node has at least one supporting node. Such considerations can be incorporated into our framework through conditions on the structure's geometry, utilizing the graph representation. Moreover, we focused on stiffness and have not taken into account secondary target properties such as, e.g., buckling loads. We agree that potential instability or failure are important considerations in practical engineering scenarios and present a valuable future extension of our framework, e.g., by integrating buckling strength⁴⁴ or compressive loading curves^{45,46} among the target properties towards the inverse design of buckling-resistant structures. Our generative modeling framework based on graph representations of truss structures can be adapted to incorporate specific design considerations such as manufacturability requirements, making it a flexible framework that can be tailored to meet various application goals. In addition, one advantage of our approach is that – when identifying an optimal structure – the latent space provides many structures with similar properties (corresponding to nearby points in the latent space), so that secondary targets can also be implemented by choosing from a selection of potential trusses with different architectures yet similar effective stiffness. We have added a comment in this direction in Section 1 of the revised Supplementary Information.

Reviewer 5:

The papers presents a graph-based deep learning generative framework to explore an extremely large design space for truss lattices. The method combines existing generative machine learning models, a variational autoencoder and an adversarial networks, which have been already successfully used to design metamaterials. Graph-based learning has been demonstrated successful across several domains including polycrystalline materials due to the capacity to model interrelations in irregular domains. The graphical representation used by the authors in this paper encode the geometric descriptors defining the truss lattice, such as strut coordinates, volume fraction, nodal connectivity, and strut cross-section, into a data structure that can in turn be used to optimize truss lattices with unusual mechanical properties on target.

While I certainly find the contents of this work scientifically solid, I also find striking similarities to ref 47, a recent paper published by some of the authors. The choice of the fundamental unit cell with cubic symmetry, the machine learning scheme, the inverse design approach, the computation of the homogenization elastic properties, and the formulation of the optimization problem to tune the elastic properties. The difference seems to lie mainly on the choice of the generative model for the truss descriptors. In addition, the generative frameworks does rely on existing models which have been already successfully applied in other domains.

Given the above, I don't see sufficient ground to warrant publication in this journal, although the technical work is sound.

Response 5: We appreciate the reviewer’s assessment that our work is scientifically sound. When it comes to its novelty and, in particular, a comparison with ref.⁹ (“ref. 47” as referred to by the reviewer) we strongly disagree, as the present study makes distinct new contributions in the following aspects:

1. **Orders-of-magnitude larger topological design space:** The truss design space presented in this work is by several orders of magnitude larger and fundamentally different from that of ref.⁹. While both works start with a set of cubic unit cells as elementary lattices, ref.⁹ extended the design space by varying the unit cell tessellation and relative density, resulting in a truss design space including 262 unique topologies. In other words, the inverse design in ref.⁹ could choose from a catalog of 262 topologies. In contrast, the topological design space in this study emerges from simultaneously varying the vertex positions and connectivities (adding/removing vertices and connections). The training dataset obtained from those variations contains 10,211 unique topologies. Moreover, the generative design framework is not limited to those topologies but may arbitrarily (up to well-defined constraints) add or remove vertices and connections, thus predicting previously unseen topologies when random sampling in the latent space. Theoretically, there is no limit to the number of unique structures you can achieve. Therefore, the design space is not only vastly larger but also fundamentally different than in ref.⁹ (and all other prior studies).
2. **A new graph-theoretic approach:** As any truss topology can be naturally represented by a graph, our graph-based treatment of truss lattices offers a significant advantage and encompasses a considerably wider design space, as explained above – not relying on an ad-hoc catalog of lattices with limited finite-dimensional design parameterizations such as that of ref.⁹ and ref.¹⁴ and all prior studies on truss metamaterials.
3. **Learning a latent design space:** In contrast to ref.⁹, which focused on the supervised learning of the forward and inverse mapping of truss geometries and properties using *discriminative* models, the *generative* modeling framework proposed in this work establishes a unified and *continuous* latent representation of a diverse truss design space with mechanical information encoded. A major advantage of generative models lies in their ability to learn the underlying distribution of the truss design space, thus enabling the generation of novel designs through the latent space, while discriminative models, as used in ref.⁹, primarily aim to learn the mapping from inputs to the target. By joint training with a property predictor, our generative model does not simply memorize the data but instead learns explicit physical relations and creates a structured latent space, in which truss lattices with similar geometries and properties are clustered in similar regions (see also Response 1.3 and Figure 3). Furthermore, the latent representation of truss lattices allows for the generation of novel structures by simple vector arithmetic operations such as interpolation and traversals, which provides a more flexible characterization of truss lattices compared with ref.⁹, which used discrete categorical design parameters. The low-dimensional, structured, regularized, property-aware, continuous, and interpretable latent space is a novelty of this study.
4. **A novel interpretable VAE approach:** The proposed VAE approach introduces a novelty in terms of distinct latent space dimensions for topology, geometry, and combined topology-geometry information of truss lattices. Quoting Reviewer #2, “*It is quite innovative to decompose the latent space dimensions into topology, geometry, and their shared dimensions*”. This decomposition enables independent and interpretable control over the design of either topological, geometrical, or both aspects of truss lattices – as discussed in the Section *Generative modeling framework* of the main article and in Section 3.2 of the revised Supplementary Information.
5. **Unprecedented extrapolation beyond the training data:** While ref.⁹ demonstrated the efficiency of their inverse design approach in predicting truss designs that match given target stiffness, those designs were well inside the sphere of the training data (due to the fact that the finite-dimensional design parameters are both lower- and upper-bounded, both sampled and predicted within those bounds). Our present work pushes the boundaries of the achievable property space by **extrapolating** far beyond the training domain. By leveraging our generative modeling framework, the graph-theoretic approach, and the unified latent representation, we design and discover truss lattices with novel properties that are significantly outside the domain of the training data. We refer the reviewer to Figure 4 in the main article for examples of such extrapolation and also reproduced it here in Figure 14 for convenience.
6. **Generalization to nonlinear mechanical responses:** In contrast to ref.⁹, which only focused on the inverse design with a target linear response, our framework showcases remarkable generalization to the nonlinear regime for the design of truss metamaterials with a target nonlinear stress-strain behavior, as shown in Figure 5

of the revised article (reproduced here in Figure 1). By training the property predictor on stress-strain curves (specifically on a finite number of discrete stress-strain pairs along the loading curve) and leveraging the mechanics-informed continuous latent representation, we can effectively guide the design of periodic trusses with desired nonlinear responses, which opens up new avenues for designing advanced metamaterials with tailored mechanical behavior for applications extending far beyond optimized stiffness (from soft robotics to impact mitigation).

Figure 14. Inverse-designed truss metamaterials based on gradient optimization: (a) maximizing Young's modulus E_{22} , (b) minimizing Poisson's ratio ν_{21} , and (c) maximizing the bulk-to-shear-modulus ratio K_V/G_V of truss lattices. Each example shows the property evolution vs. the number of optimization iteration steps, including a few selected structures at the indicated points, as well as the property path compared to the training dataset in the relevant property spaces (each dot represents a truss in the training data, which is color-coded by the radius of the beams with circular cross-section).

On the use of VAEs for the design of metamaterials: We agree with the reviewer that VAEs and GANs have already been used for the design of metamaterials. However, not all metamaterials design spaces rate on the same level of difficulty; therefore, the use of ML methods such as VAE for a much simpler metamaterial does not close the field (not to mention graph-based VAEs which are the novel focus here). Prominent works such as refs.^{47–50} were limited to low-dimensional Euclidean parameterizations (sometimes as low as 5–10 parameters, making the use of VAEs/GANs unnecessary or redundant) or image-based 2D pixelated composites (which are not suitable for practical applications). Truss lattices – the most popular and easily manufacturable class of architected materials – have slender beams and very low relative densities that preclude any image-based approach and usually admit only non-Euclidean graph-based representations. Despite being at the forefront of engineering applications among all metamaterials, a systematic design approach (including our previous work ref.⁹) for truss metamaterials has not yet been presented for the above reasons. This work is a first step in the direction to address this knowledge and technology gap.

References

1. Ross, E. & Hambleton, D. Using graph neural networks to approximate mechanical response on 3D lattice structures. *Proc. AAG2020-Advances Archit. Geom.* **24**, 466–485 (2021).
2. Meyer, P. P., Bonatti, C., Tancogne-Dejean, T. & Mohr, D. Graph-based metamaterials: Deep learning of structure-property relations. *Mater. & Des.* **223**, 111175 (2022).
3. Indurkar, P. P., Karlapati, S., Shaikeea, A. J. D. & Deshpande, V. S. Predicting deformation mechanisms in architected metamaterials using gnn. *arXiv preprint arXiv:2202.09427* (2022).
4. Gómez-Bombarelli, R. *et al.* Automatic chemical design using a data-driven continuous representation of molecules. *ACS Cent. Sci.* **4**, 268–276 (2018).
5. Ingraham, J., Garg, V., Barzilay, R. & Jaakkola, T. Generative models for graph-based protein design. *Adv. neural information processing systems* **32** (2019).
6. Panetta, J. *et al.* Elastic textures for additive fabrication. *ACM Transactions on Graph. (TOG)* **34** (2015).
7. Wang, L., Tao, S., Zhu, P. & Chen, W. Data-driven topology optimization with multiclass microstructures using latent variable gaussian process. *J. Mech. Des.* **143** (2021).
8. Wang, C. *et al.* Concurrent design of hierarchical structures with three-dimensional parameterized lattice microstructures for additive manufacturing. *Struct. Multidiscip. Optim.* **61**, 869–894 (2020).
9. Bastek, J.-H., Kumar, S., Telgen, B., Glaesener, R. N. & Kochmann, D. M. Inverting the structure–property map of truss metamaterials by deep learning. *Proc. Natl. Acad. Sci.* **119**, e2111505119 (2022).
10. Xu, S., Shen, J., Zhou, S., Huang, X. & Xie, Y. M. Design of lattice structures with controlled anisotropy. *Mater. & Des.* **93**, 443–447 (2016).
11. Watts, S., Arrighi, W., Kudo, J., Tortorelli, D. A. & White, D. A. Simple, accurate surrogate models of the elastic response of three-dimensional open truss micro-architectures with applications to multiscale topology design. *Struct. Multidiscip. Optim.* **60**, 1887–1920 (2019).
12. Lipton, Z. C. The mythos of model interpretability: In machine learning, the concept of interpretability is both important and slippery. *Queue* **16**, 31–57 (2018).
13. Montero, M. L., Ludwig, C. J., Costa, R. P., Malhotra, G. & Bowers, J. The role of disentanglement in generalisation. In *International Conference on Learning Representations* (2021).
14. Fernández, M., Fritzen, F. & Weeger, O. Material modeling for parametric, anisotropic finite strain hyperelasticity based on machine learning with application in optimization of metamaterials. *Int. J. for Numer. Methods Eng.* **123**, 577–609 (2022).
15. Xu, K., Huang, D. Z. & Darve, E. Learning constitutive relations using symmetric positive definite neural networks. *J. Comput. Phys.* **428**, 110072 (2021).
16. Jekel, C. F., Swartz, K. E., White, D. A., Tortorelli, D. A. & Watts, S. E. Neural network layers for prediction of positive definite elastic stiffness tensors. *arXiv preprint arXiv:2203.13938* (2022).

17. Vlassis, N. N., Ma, R. & Sun, W. Geometric deep learning for computational mechanics part i: anisotropic hyperelasticity. *Comput. Methods Appl. Mech. Eng.* **371**, 113299 (2020).
18. Yamaguchi, K., Yasuda, H., Tsujikawa, K., Kunimine, T. & Yang, J. Graph-theoretic estimation of reconfigurability in origami-based metamaterials. *Mater. & Des.* **213**, 110343 (2022).
19. Kollmann, H. T., Abueidda, D. W., Koric, S., Guleryuz, E. & Sobh, N. A. Deep learning for topology optimization of 2d metamaterials. *Mater. & Des.* **196**, 109098 (2020).
20. Chen, Z., Ogren, A., Daraio, C., Brinson, L. C. & Rudin, C. How to see hidden patterns in metamaterials with interpretable machine learning. *Extrem. Mech. Lett.* **57**, 101895 (2022).
21. Kumar, S., Tan, S., Zheng, L. & Kochmann, D. M. Inverse-designed spinodoid metamaterials. *npj Comput. Mater.* **6**, 73 (2020).
22. Alderete, N. A., Pathak, N. & Espinosa, H. D. Machine learning assisted design of shape-programmable 3d kirigami metamaterials. *npj Comput. Mater.* **8**, 191 (2022).
23. Li, X. *et al.* Designing phononic crystal with anticipated band gap through a deep learning based data-driven method. *Comput. Methods Appl. Mech. Eng.* **361**, 112737 (2020).
24. Jiang, W. *et al.* Dispersion relation prediction and structure inverse design of elastic metamaterials via deep learning. *Mater. Today Phys.* **22**, 100616 (2022).
25. Capela, F., Nouchi, V., Van Deursen, R., Tetko, I. V. & Godin, G. Multitask learning on graph neural networks applied to molecular property predictions. *arXiv preprint arXiv:1910.13124* (2019).
26. Pasini, M. L., Zhang, P., Reeve, S. T. & Choi, J. Y. Multi-task graph neural networks for simultaneous prediction of global and atomic properties in ferromagnetic systems. *Mach. Learn. Sci. Technol.* **3**, 025007 (2022).
27. Sanyal, S. *et al.* Mt-cgcn: Integrating crystal graph convolutional neural network with multitask learning for material property prediction. *arXiv preprint arXiv:1811.05660* (2018).
28. Sanchez-Lengeling, B. & Aspuru-Guzik, A. Inverse molecular design using machine learning: Generative models for matter engineering. *Science* **361**, 360–365 (2018).
29. Simonovsky, M. & Komodakis, N. Graphvae: Towards generation of small graphs using variational autoencoders. In *Artificial Neural Networks and Machine Learning–ICANN 2018: 27th International Conference on Artificial Neural Networks, Rhodes, Greece, October 4–7, 2018, Proceedings, Part I 27*, 412–422 (Springer, 2018).
30. You, J., Ying, R., Ren, X., Hamilton, W. & Leskovec, J. Graphrnn: Generating realistic graphs with deep auto-regressive models. In *International conference on machine learning*, 5708–5717 (PMLR, 2018).
31. Satorras, V. G., Hoogeboom, E. & Welling, M. E (n) equivariant graph neural networks. In *International conference on machine learning*, 9323–9332 (PMLR, 2021).
32. Reiser, P. *et al.* Graph neural networks for materials science and chemistry. *Commun. Mater.* **3**, 93 (2022).
33. Han, J., Huang, W., Xu, T. & Rong, Y. Equivariant graph hierarchy-based neural networks. *Adv. Neural Inf. Process. Syst.* **35**, 9176–9187 (2022).
34. Duan, T. & Lee, J. Graph embedding vae: A permutation invariant model of graph structure. *arXiv preprint arXiv:1910.08057* (2019).
35. Niu, C. *et al.* Permutation invariant graph generation via score-based generative modeling. In *International Conference on Artificial Intelligence and Statistics*, 4474–4484 (PMLR, 2020).
36. Winter, R., Noé, F. & Clevert, D.-A. Permutation-invariant variational autoencoder for graph-level representation learning. *Adv. Neural Inf. Process. Syst.* **34**, 9559–9573 (2021).
37. Batzner, S. *et al.* E (3)-equivariant graph neural networks for data-efficient and accurate interatomic potentials. *Nat. communications* **13**, 2453 (2022).
38. Jørgensen, P. B. & Bhowmik, A. Equivariant graph neural networks for fast electron density estimation of molecules, liquids, and solids. *npj Comput. Mater.* **8**, 183 (2022).

39. White, T. Sampling generative networks. *arXiv preprint arXiv:1609.04468* (2016).
40. Vlassis, N. N. & Sun, W. Geometric learning for computational mechanics part ii: Graph embedding for interpretable multiscale plasticity. *Comput. Methods Appl. Mech. Eng.* **404**, 115768 (2023).
41. Thomas, N. *et al.* Tensor field networks: Rotation-and translation-equivariant neural networks for 3d point clouds. *arXiv preprint arXiv:1802.08219* (2018).
42. Fuchs, F., Worrall, D., Fischer, V. & Welling, M. Se (3)-transformers: 3d roto-translation equivariant attention networks. *Adv. Neural Inf. Process. Syst.* **33**, 1970–1981 (2020).
43. Cai, C. *et al.* Equivariant geometric learning for digital rock physics: estimating formation factor and effective permeability tensors from morse graph. *Int. J. for Multiscale Comput. Eng.* **21** (2023).
44. Maurizi, M., Gao, C. & Berto, F. Inverse design of truss lattice materials with superior buckling resistance. *npj Comput. Mater.* **8**, 247 (2022).
45. Wang, Y., Zeng, Q., Wang, J., Li, Y. & Fang, D. Inverse design of shell-based mechanical metamaterial with customized loading curves based on machine learning and genetic algorithm. *Comput. Methods Appl. Mech. Eng.* **401**, 115571 (2022).
46. Deng, B. *et al.* Inverse design of mechanical metamaterials with target nonlinear response via a neural accelerated evolution strategy. *Adv. Mater.* **34**, 2206238 (2022).
47. Wang, L. *et al.* Deep generative modeling for mechanistic-based learning and design of metamaterial systems. *Comput. Methods Appl. Mech. Eng.* **372**, 113377 (2020).
48. Xue, T., Wallin, T. J., Menguc, Y., Adriaenssens, S. & Chiaramonte, M. Machine learning generative models for automatic design of multi-material 3d printed composite solids. *Extrem. Mech. Lett.* **41**, 100992 (2020).
49. Chen, C.-T. & Gu, G. X. Generative deep neural networks for inverse materials design using backpropagation and active learning. *Adv. Sci.* **7**, 1902607 (2020).
50. Mao, Y., He, Q. & Zhao, X. Designing complex architected materials with generative adversarial networks. *Sci. advances* **6**, eaaz4169 (2020).

REVIEWERS' COMMENTS

Reviewer #1 (Remarks to the Author):

The authors have provided a thorough revision of their manuscript and rebuttal to all reviewer comments. The novelty of the present work compared to previous works has been clearly highlighted and thus I recommend acceptance of the manuscript in its present form.

Reviewer #2 (Remarks to the Author):

The authors have diligently addressed all the comments and substantially enhanced the manuscript by incorporating new calculations and additional text. They have now demonstrated the inverse design of trusses with customized mechanical properties in "nonlinear regimes", a significant advancement beyond the originally presented linear regime. This novel development significantly enhances the manuscript and broadens its applicability to a range of materials, making it suitable for publication in Nature Communications.

Reviewer #3 (Remarks to the Author):

Very nice and improved work. Well done.

Therefore, the revised manuscript can be considered for publication.

Reviewer #4 (Remarks to the Author):

The authors have carefully revised their manuscript and well addressed my concerns. The quality of the revised manuscript has been significantly improved.

Thus, this paper can be accepted.

Reviewer #5 (Remarks to the Author):

The authors have explained in the depth the distinct features of this work over the existing literature including their recent paper which shares some basic aspects. I appreciate the inclusion of the section on the non-linear mechanical properties which demonstrates its application beyond the linear regime.

The other additions in response to the other reviewers' comments also contribute to increase the quality of this work.